# Antibodies from dengue patients with prior exposure to Japanese encephalitis virus are broadly neutralizing against Zika virus

Gielenny M. Salem [1,12], Jedhan Ucat Galula[1,12], Shang-Rung Wu[2,3,13], Jyung-Hurng Liu [4,13], Yen-Hsu Chen [5,6,7], Wen-Hung Wang[5,6,7], Sheng-Fan Wang[6,8], Cheng-Sheng Song[1], Fan-Chi Chen[9], Adrian B. Abarientos[1], Guan-Wen Chen[2], Cheng-I Wang[10] & Day-Yu Chao [1,9,11✉]

Exposure to multiple mosquito-borne flaviviruses within a lifetime is not uncommon; however, how sequential exposures to different flaviviruses shape the cross-reactive humoral response against an antigen from a different serocomplex has yet to be explored. Here, we report that dengue-infected individuals initially primed with the Japanese encephalitis virus (JEV) showed broad, highly neutralizing potencies against Zika virus (ZIKV). We also identified a rare class of ZIKV-cross-reactive human monoclonal antibodies with increased somatic hypermutation and broad neutralization against multiple flaviviruses. One huMAb, K8b, binds quaternary epitopes with heavy and light chains separately interacting with overlapping envelope protein dimer units spanning domains I, II, and III through cryo-electron microscopy and structure-based mutagenesis. JEV virus-like particle immunization in mice further confirmed that such cross-reactive antibodies, mainly IgG3 isotype, can be induced and proliferate through heterologous dengue virus (DENV) serotype 2 virus-like particle stimulation. Our findings highlight the role of prior immunity in JEV and DENV in shaping the breadth of humoral response and provide insights for future vaccination strategies in flavivirus-endemic countries.

[1] Graduate Institute of Microbiology and Public Health, College of Veterinary Medicine, National Chung Hsing University, Taichung City 402, Taiwan. [2] Institute of Oral Medicine, School of Dentistry, College of Medicine, National Cheng Kung University, Tainan City 701, Taiwan. [3] Institute of Basic Medical Sciences, College of Medicine, National Cheng Kung University, Tainan City 701, Taiwan. [4] Graduate Institute of Genomics and Bioinformatics, College of Life Sciences, National Chung Hsing University, Taichung City 40227, Taiwan. [5] School of Medicine, College of Medicine, National Sun Yat-Sen University, Kaohsiung City 80424, Taiwan. [6] Center for Tropical Medicine and Infectious Disease Research, Kaohsiung Medical University, Kaohsiung City 80708, Taiwan. [7] Division of Infectious Diseases, Department of Internal Medicine, Kaohsiung Medical University Hospital, Kaohsiung Medical University, Kaohsiung City 80708, Taiwan. [8] Department of Medical Laboratory Science and Biotechnology, Kaohsiung Medical University, Kaohsiung City 80708, Taiwan. [9] Doctoral Program in Microbial Genomics, National Chung Hsing University and Academia Sinica, Taichung City 402, Taiwan. [10] Singapore Immunology Network, Agency for Science, Technology and Research (A*STAR), 8A Biomedical Grove, Immunos, Singapore 138648, Singapore. [11] Department of Post-Baccalaureate Medicine, College of Medicine, National Chung Hsing University, Taichung City 402, Taiwan. [12] These authors contributed equally: Gielenny M. Salem, Jedhan Ucat Galula. [13] These authors jointly supervised this work: Shang-Rung Wu, Jyung-Hurng Liu. ✉email: dychao@nchu.edu.tw

Flaviviruses, the largest member of the family *Flaviviridae*, present a global threat to public health[1]. Many of the mosquito-borne flaviviruses are established human pathogens, including the Japanese encephalitis virus (JEV), Yellow fever virus (YFV), West Nile virus (WNV), the four serotypes of dengue virus (DENV), and the re-emerged Zika virus (ZIKV)[2]. Co-circulation of multiple flaviviruses in the same geographic locations, combined with human global mobility and travel vaccine coverage for YFV and JEV, have increased the likelihood of exposure to multiple flaviviruses within a lifetime[3,4]. However, how pre-existing immunity will influence the antibody response upon subsequent infections or vaccination with heterologous flaviviruses remains poorly understood.

Flaviviruses are enveloped viruses containing a single-strand, positive-sense RNA with an 11-Kb genome and encapsidated by three structural proteins, namely the capsid (C), pre-membrane/membrane (prM/M), and envelope (E) proteins. The immune response to flaviviral infection mainly targets the E proteins and is known to be dominated by transient and highly cross-reactive (CR) antibodies during the acute and early convalescent phase. In the late convalescent phase, the immune response generates type-specific neutralizing antibodies but lacks durable and high cross-neutralizing antibodies against viruses from different serotypes or serocomplexes[5–7]. The envelope dimer epitope (EDE) human monoclonal antibodies (huMAbs), which broadly neutralize the four dengue virus serotypes by recognizing the quaternary epitopes on the virion surface, have been mostly isolated from individuals exposed to secondary dengue infections[8]. Recurring huMAbs CR to DENV and ZIKV have also been reported[9,10] from the regions where both viruses co-circulated. However, the antibody profiles of individuals residing in areas where JEV and DENV co-circulated have not been investigated in detail.

Here, we used a unique Taiwan cohort to gain insight into how DENV infection in individuals with pre-existing JEV immunity shapes neutralizing antibody responses against multiple flaviviruses through interdisciplinary and complementary approaches, including epidemiology, immunology, structural biology, and animal studies. The results will guide future vaccination strategies, leading to the generation of broadly neutralizing antibodies against different flavivirus serocomplexes.

## Results

### High ZIKV neutralization titers among dengue-infected individuals with pre-existing JEV immunity

Based on the nationwide surveillance system established by the Taiwan Centers for Disease Control, only imported ZIKV cases were detected without local transmission[11–13]. In contrast, JEV is endemic on the island; thus, a nationwide JEV pediatric vaccination program has been implemented since 1968[14,15]. Periodic DENV epidemics have also occurred in Taiwan since 1981, with the largest outbreaks recorded in 2014–2015 in the southern part of the island[16,17]. To investigate the antibody profile of DENV-infected patients to different flaviviruses, including ZIKV, plasma of healthy volunteers and DENV-infected febrile (DF) patients were tested for the presence of DENV, JEV, and ZIKV-neutralizing antibodies using 90% foci reduction as the cut-off (FRμNT90) (Fig. 1a). Expectedly, DF-confirmed samples had significantly elevated DENV FRμNT90 titers than the healthy donors. Only four healthy individuals showed DENV FRμNT90 titers between 50 and 80, with one donor (1413) showing DENV-1, -2, and -3 neutralizing titers, suggesting previous dengue infection or potential asymptomatic dengue infection given the overlapping residential area of the donors in southern Taiwan. Significantly elevated JEV FRμNT90 titers among the DF patients were also observed compared to the healthy controls (Fig. 1a), although thirteen healthy donors from the older age group showed varied JEV-neutralizing titers (Supplementary Fig. 1a). Surprisingly, the anti-ZIKV-neutralizing titers were substantially higher among DF patients than the healthy individuals (Fig. 1a).

To distinguish between individuals immune to JEV due to vaccination or potential natural infection, the study population was divided into two age groups: 20–30 years old (post-JEV vaccination program) and 50–70 years old (pre-JEV vaccination program). Among DF patients, the JEV and ZIKV FRμNT90 geometric mean titers (GMTs) were statistically higher in the 50–70 age group (JEV = 251; ZIKV = 125) than in the 20–30 age group (JEV = 29; ZIKV = 21) (Fig. 1b). Similar antibody neutralization kinetics and titers were observed against other global or Taiwan clinical DENV isolates or the Asian and African ZIKV strains regardless of the age group (Supplementary Fig. 1b, c). None of the donors showed neutralization against YFV (Supplementary Fig. 1d).

We further classified the donors to determine the impact of JEV and DENV immunity on ZIKV neutralization. Individuals from the JEV$^{pos}$DENV$^{high}$ group had significantly elevated circulating ZIKV-neutralizing antibodies (GMT = 84) than the others (GMT < 10) for both age groups (Fig. 1c and Supplementary Fig. 1e, f). Among the DF individuals with elevated neutralizing ZIKV titers (ZIKV$^{pos}$), 28 of the 31 subjects (90.3%) were from the JEV$^{pos}$DENV$^{high}$ group. In contrast, only three individuals (9.68%) recorded high ZIKV neutralization titers from the other groups (Fig. 1d). The percentage of individuals with high JEV and DENV titers and consequently high ZIKV titers was statistically higher than the proportion of individuals with only one immunity, suggesting that sequential exposures to JEV and DENV could potentially induce high and cross-neutralizing antibodies against ZIKV that have yet to be encountered by the host.

### ZIKV-CR huMAbs identified from a donor at late convalescence

Previous longitudinal analyses suggested that short-lived but cross-neutralizing antibody responses can be generated after heterologous DENV exposures[7,18,19] and are greatest among dengue-infected plasma samples collected during the early convalescent phase post-infection[20]. To confirm if such cross-neutralizing ZIKV antibodies persist, we successfully recalled one recovered donor (KH1891) from the same Taiwan cohort more than 18 months after the dengue infection despite the difficulty of recalling recovered donors. While JEV-neutralizing titers increased slightly, a significant decrease in DENV-1 to -4 FRμNT90 titers was observed, and ZIKV-neutralizing titers remained prominent (Fig. 2a and Supplementary Fig. 2a, P < 0.001).

Next, we utilized the peripheral blood mononuclear cells (PBMCs) from donor KH1891 to interrogate the presence of ZIKV-specific B cells. Given the low percentage of circulating memory B cells (MBCs) in the peripheral blood at late convalescence, we generated an in-house B cell sorting and co-culture strategy for ZIKV-reactive B cell clone screening to isolate huMAbs (Supplementary Fig. 2a). After single-cell sorting, 1.11% of CD19$^{+}$IgM$^{-}$IgA$^{-}$IgD$^{-}$ B cells were co-cultivated with CD40L-expressing feeder cells in a medium supplemented with interleukin-2 (IL-2) and IL-21 for two weeks[21] (Supplementary Fig. 2b–d). Following ZIKV-reactive B cell screening, human antibody variable genes were amplified from 24 ZIKV-reactive B cell clones and expressed as full-length IgG1 with native-paired heavy and light chains. For initial crude screening, we focused on broad and potent activities by ZIKV neutralization (%FRμNT50) and found that 42% (10/24) showed neutralizing activities; hence, termed as ZIKV-CR huMAbs (Supplementary Fig. 3a) and were

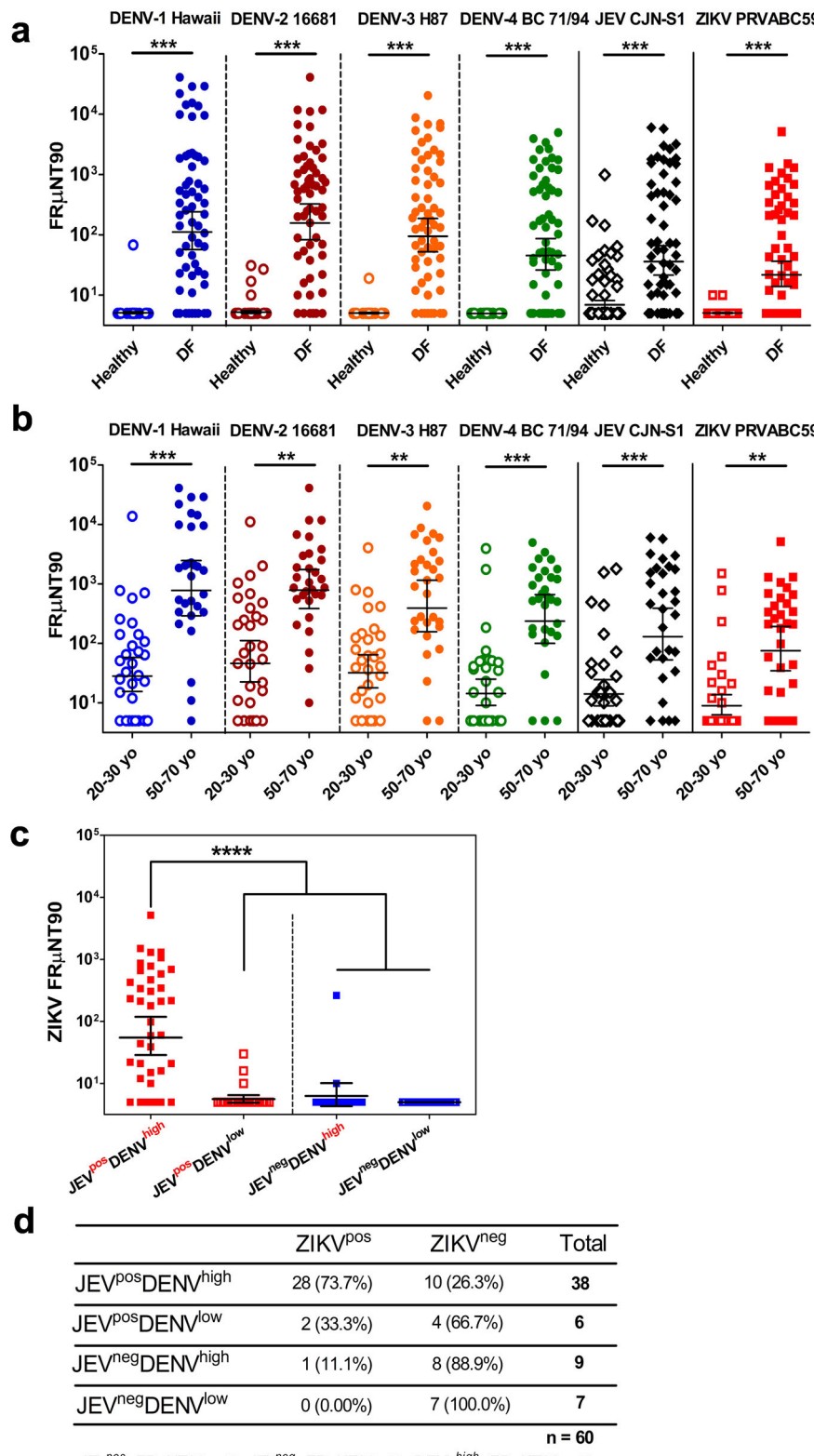

All ten ZIKV-CR huMAbs showed similar binding activities to all virus-like particles (VLPs) (Fig. 2b and Supplementary Fig. 3b), except K5, which showed lowest binding activities to all VLPs

(Fig. 2c). Of the ten, five huMAbs (K5, K8b, K12, K22, and K23) demonstrated variable yet broad and significant neutralization against three flavivirus serocomplexes; three of which (K5, K8b, and K12) showed significant half-maximal inhibitory concentrations (IC50 neutralization titers, $P < 0.05$) against DENV-1 to -4,

**Fig. 1 Dengue-immune donors with prior JEV exposure have elevated ZIKV-neutralizing titers. a** Neutralization antibody profile against the four serotypes of DENV, JEV, and ZIKV of healthy ($n = 80$) and dengue-infected febrile (DF, $n = 60$) individuals. Two-fold serially diluted plasma samples from healthy and DF individuals were evaluated for neutralization against the prototype strains of DENV-1 to -4 serotypes, JEV, and ZIKV using FRμNT90. Empty and shaded circles represent the FRμNT90 values of the healthy and DF individuals, respectively. **b** Comparison of the neutralization antibody profile of DF individuals against DENV-1 to -4 serotypes, JEV, and ZIKV prototype virions based on age classifications: 20–30 years old ($n = 30$) and 50–70 years old ($n = 30$). Empty and shaded circles represent the FRμNT90 values of the 20–30 and 50–70 age groups, respectively. **c** Assessment of the breadth of ZIKV neutralization among the four groups of donors dichotomized according to the presence (pos) or absence (neg) of protective or high neutralizing FRμNT90 titers against JEV and DENV, respectively. JEV-immune individuals with high ($JEV^{pos}DENV^{high}$; $n = 39$) or low DENV immunity ($JEV^{pos}DENV^{low}$; $n = 24$) are represented by red filled or empty squares, respectively. JEV-negative donors with high ($JEV^{neg}DENV^{high}$; $n = 12$) or low DENV ($JEV^{neg}DENV^{low}$; $n = 65$) immunity are shown in blue filled or empty squares, respectively. **d** Effect of JEV and DENV immunity in DF individuals to ZIKV-neutralizing antibody titers. Fisher's exact test was performed in $n = 60$ with a p-value of 0.000013. All data are representative of three independent experiments. In (**a–c**), the black solid horizontal bars represent the geometric mean titers (GMT) of samples with SD analyzed within each group. FRμNT90 titer <10 was represented with 5 for graphic display and statistical analysis. Significant differences were tested using one-way ANOVA followed by Tukey's multiple comparisons post-test, with levels of significance defined by ***$P < 0.0001$; **$P < 0.001$. Each point shows the mean of data from two independent experiments. DENV dengue virus, JEV Japanese encephalitis virus, ZIKV Zika virus, FRμNT focus-reduction microneutralization test, VLP virus-like particle, ANOVA analysis of variance, SD standard deviation.

JEV, and ZIKV while two huMAbs (K22 and K23) displayed potent activities against selected DENV serotypes, JEV, and ZIKV (Fig. 2d and Supplementary Fig. 3c). In contrast, the other five huMAbs (K7, K8, K9, K11, and K15) have restricted neutralization to DENV-1 to -4 and JEV, with K11 and K15 displaying significant IC50 titers to DENV-1 to -4 and JEV (Supplementary Fig. 3c). ZIKV-CR huMAbs showed a broad range of neutralization potency (IC50 from 0.180 to 10.0 μg/mL) (Fig. 2e). Interestingly, K8b and K5, respectively, displayed the broadest and most potent neutralization (FRμNT50) against ZIKV (0.180 μg/mL; 0.249 μg/mL), DENV-2 (0.207 μg/mL; 0.215 μg/mL), DENV-4 (0.356 μg/mL), JEV (0.523 μg/mL; 0.682 μg/mL), and DENV-1 (0.618 μg/mL; 0.771 μg/mL), but moderate neutralizing activity to DENV-3 (2.60 μg/mL; 1.208 μg/mL) and against DENV-4 (1.513 μg/mL) by K5 (Fig. 2f, g).

Further investigating the gene usage from the isolated huMAbs, the MBCs expressing the ZIKV-CR huMAbs utilized gene segments IGHV 1-69 (K5, K7, K8, K12, K15, K22, and K23) and IGHV 1–2 (K8b, K9, and K11) (Fig. 3a, b), compared to the non-ZIKV-CR huMAbs with broader IGHV gene assignment (Supplementary Fig. 4a, b). We also observed identical V, D, J, and heavy chain complementarity determining region 3 (HCDR3) sequences among ZIKV-CR huMAbs (Fig. 3b), suggesting a close clonal relationship of B cells expressing the broadly neutralizing antibodies[22]. Despite heavy chain similarities, the light chain V-genes of the ZIKV-CR huMAbs are encoded by diverse germline genes, such as IGκV1, IGκV3, and IGκV4, similar to non-ZIKV-CR huMAbs (Supplementary Fig. 4c, d), and consequently, diverse kappa light chain CDR3 (κCDR3) sequences (Fig. 3c, d). High somatic hypermutation (SHM) rates among the ZIKV-CR huMAbs in either heavy or light chain V-genes were also observed (Fig. 3e), equivalent to secondary dengue infections and influenza vaccination or infection[23] and suggestive of enhanced affinity maturation[24,25]; thus, higher relative proportion (~80%) of dissimilar amino acids (Supplementary Fig. 4e). On average, the ZIKV-CR huMAbs accumulated more non-silent nucleotide mutations in the heavy chain ($n = 29$) than the non-ZIKV-CR huMAbs ($n = 20$) in this study and previously known influenza-specific MBCs ($n = 20$)[23], with significant mutations demonstrated in HCDR2 and HFR3 domains (Supplementary Fig. 4f, g). Although our study only included one donor, the average number of V-gene nucleotide mutations in ZIKV-CR huMAbs of KH1891 was comparable to the heavy chain V-gene mutations reported among donors with high ZIKV-neutralizing titers[9]. Six of the ZIKV-CR huMAbs (K5, K8b, K9, K11, K22, and K23) carried the increased SHM (>20%) in either the heavy or light chain or both Ig genes, while four showed minimal changes

compared to the germline sequences (Fig. 3f). Compared to the non-ZIKV-CR huMAbs with lower SHM (Supplementary Fig. 4h–j), ZIKV-CR huMAbs showed unique heavy and light chain V(D)J rearrangements and pairing. Taken together, the ZIKV-CR neutralizing huMAbs, including K8b and K5, were isolated from the peripheral blood at the late convalescent stage and showed high degrees of non-silent SHM, suggesting the persistence and continuing maturation of the MBCs upon repeated exposures to JEV and DENV.

**Structure-guided epitope mapping of huMAbs K8b and K5**. We chose K8b for epitope characterization due to its broad neutralization activities and higher expression level. Instead of performing random shotgun mutation on the complete E gene, we developed a structure-based strategy using a previously generated mature form virus-like particles of dengue virus serotype 2 (mD2VLP)[26] to form an immune complex with K8b-IgG1 (Fig. 4a–c). The cryo-EM micrographs of the mD2VLP-K8b-IgG1 immune complex (Supplementary Fig. 5a) revealed the heterogeneity in particle size, which we attributed to the variability of mD2VLPs[26]. The inherent heterogeneity of mD2VLPs and antibody hinges made improving structure resolution challenging. Nevertheless, the structure of the mD2VLP-IgG1 immune complex with unresolved Fc was determined (Fig. 4c). The resolution of this structure was calculated to be 16.3 Å (Supplementary Fig. 5a). Details of the cryo-EM data collection, refinement, and validation statistics were shown in Table 1. The surface-rendered structure at a high threshold (1.2 σ) clearly demonstrated the separation of variable domains from constant domain within the density map, greatly assisting in the orientation of the F(ab) fragments (Fig. 4d). A more precise fitting was achieved using HEX, which calculated the binding energy and determined the most probable binding orientation of the F(ab) fragment with another molecule (see Methods). To elucidate how the F(ab) fragments were coupled, a more permissive mask filtering approach was implemented during the reconstruction process (Supplementary Fig. 5b). Notably, the configuration of the Fc region dictated the K8b-IgG1 bivalent arms, which bound separately to different E-dimer units, suggesting that K8b-IgG1 may function not only as an interdomain antibody but also an inter-dimer antibody (Fig. 4d).

The resolution of the K8b-IgG1 immune complex was further enhanced to 12 Å using MODELLER to map the potential interacting residues between the E proteins on an mD2VLP and the heavy or light chain of K8b-IgG1 (Fig. 4e, f). Based on the predicted interacting residues on the E protein of DENV-2, seven residues spanning across domains I, II, and III were mutated into

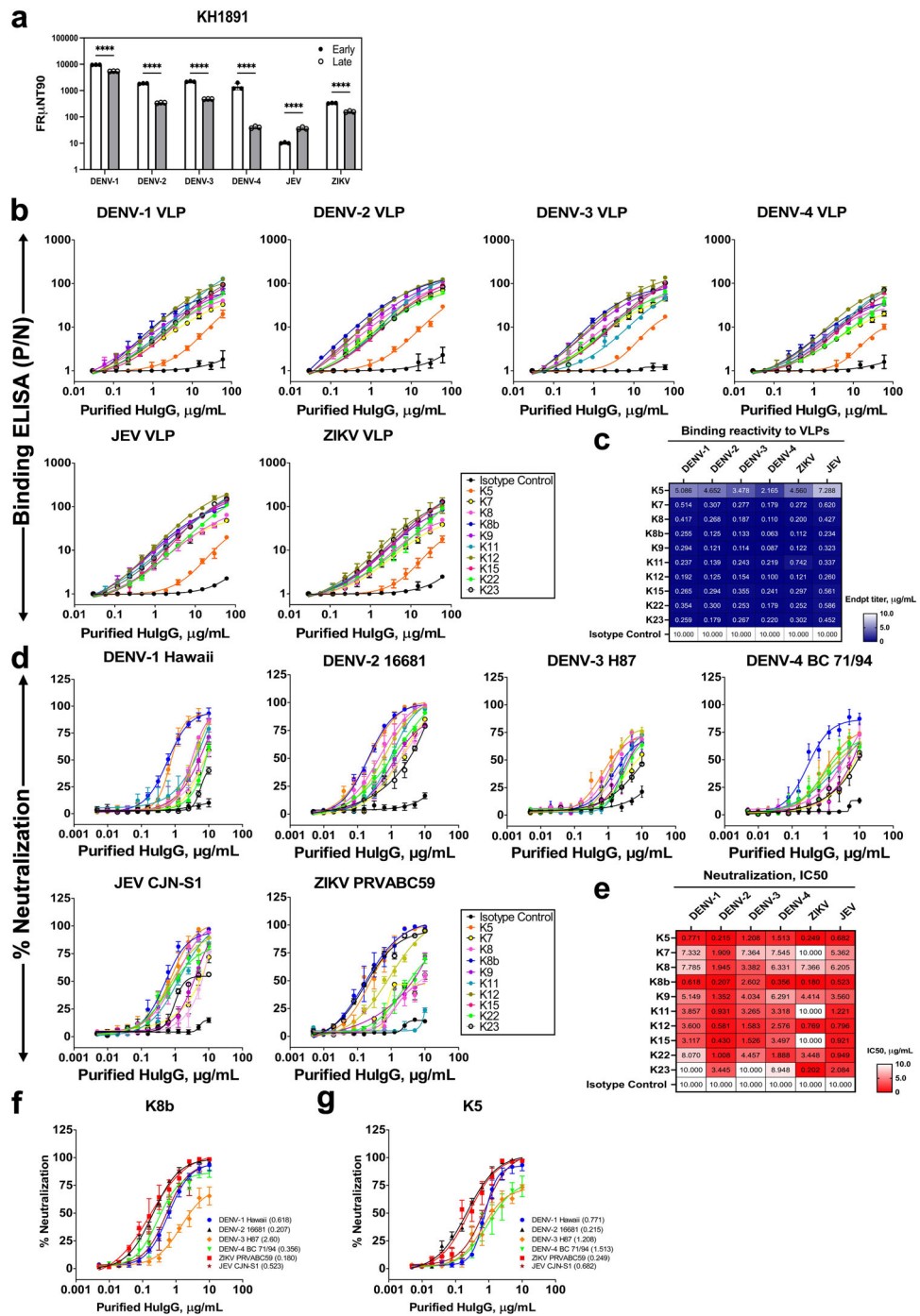

alanine on ZIKV VLP based on the criteria described in Methods (see SDM and epitope mapping experiments section), such as R73, T76, and D87 on the bc loop of domain I, R193 on domain I and II linker region, T231 on the hi loop of domain II, G302 on domain I and III linker region, and T366 on DE loop of domain III (Fig. 4g and Supplementary Fig. 6a, b). The recognition of K8b-IgG1 to D87A, R193A, T231A, G302A, and T366A mutant antigens were significantly reduced by as much as 50% relative to ZIKV WT VLP (Fig. 4h). In contrast, K8b and K5 binding to residue 101-mutated VLPs were unaffected (Supplementary Fig. 6d–g). The overall binding reactivities of K5-IgG1 to most mutant ZIKV VLPs were similar to K8b-IgG1 except for two residues, ZIKV R193A and ZIKV T231A. Mutation of R193 and

T231 showed abrogation of K5 binding by 70% (Fig. 4i), indicating substantial functional contribution of both residues to K5 and ZIKV interactions.

To validate if antibody geometry could affect the binding and avidity to putative interdomain or inter-dimer epitopes on different E dimers, the full-length bivalent IgG1 of K8b (K8b-IgG1) was engineered and expressed as bivalent K8b-F(ab')$_2$ and monovalent K8b-F(ab) (Fig. 5a–d). Parallel binding and neutralization experiments were performed on K8b-IgG1, K8b-F(ab')$_2$, and monovalent K8b-F(ab) against ZIKV, JEV, and the four DENV serotypes. Bivalent K8b-IgG1 and K8b-F(ab')$_2$ have similar neutralization against DENV-1, DENV-2, DENV-4, ZIKV, and JEV (FRμNT50: 7-17 nM), but K8b-F(ab')$_2$ showed

**Fig. 2 Functional characterization of anti-ZIKV human monoclonal antibodies (huMAbs) from donor KH1891. a** Neutralizing antibody titers of donor KH1891 against the prototype strains of the DENV serotypes 1–4, JEV, and ZIKV at less than 1 month and 18 months post-infection plasma collection, representing the early and late convalescent stages, respectively. Significant differences were tested using two-way ANOVA followed by Bonferroni post-tests from three independent experiments. ****$P < 0.001$. **b–e** Characterization of purified huMAbs, expressed as IgG1, isolated from donor KH1891. **b** The binding reactivity against virus-like particles (VLPs) and **d** in vitro neutralization activities of 10 purified huMAbs against DENV-1 to -4, JEV, and ZIKV were measured by ELISA and FRμNT, respectively. Each VLP was properly titrated to obtain equimolar concentrations per well before use in antigen-capture ELISAs. Each point shows the mean ± SD of data from two and three independent experiments for binding ELISA and microneutralization assays, respectively. A nonlinear curve was generated for all antibody dilution series for all of the assays. The corresponding antibody identities are shown in the box. **c, e** Heatmap of VLP-binding profile (shades of blue) (**c**) and of neutralization profile (shades of red) (**e**) of the 11 huMAbs against recombinantly produced DENV-1 to -4, JEV, and ZIKV VLPs and prototype strains of DENVs, JEV, and ZIKV virus particles, respectively. The heatmaps were generated based on the minimal binding concentration (or endpoint titer values) against each VLP and the IC50 values for each virion; exact numerical values expressed in μg/mL are shown in each box. Each huMAb was tested in duplicates for each of the binding ELISA assays, while FRμNT data were representative of three independent experiments. A purified human IgG1 was used as an isotype control in all assays. Data are representative of three independent experiments. **f, g** FRμNT50 (μg/mL) values of K8b (**f**) and K5 (**g**) for neutralization of the prototype strains of the four DENV serotypes, JEV, and ZIKV. Each point shows the mean ± SD of data from three independent experiments. The geometric mean inhibitory concentration (μg/mL) at 50% (IC50) is indicated in parenthesis. DENV dengue virus, JEV Japanese encephalitis virus, ZIKV Zika virus, ELISA enzyme-linked immunosorbent assay, FRμNT focus-reduction microneutralization test, VLP virus-like particle, ANOVA analysis of variance, SD standard deviation.

a slight improvement in neutralization against DENV-3 (FRμNT50: 26-50 nM) (Fig. 5e). Although both VLP and virions form E-dimer on the surface of the icosahedral sphere, the observed VLP binding kinetics for K8b-IgG1 and K8b-F(ab')$_2$ were consistent with neutralization for DENV-2 and ZIKV only but not for JEV, DENV-1, DENV-3, and DENV-4 VLPs probably due to the difference in geometry between VLP ($T = 1$) and virion ($T = 3$) (Fig. 5f). Unlike K8b-F(ab')$_2$, significantly reduced neutralizing activity to all six viruses was observed for K8b-F(ab) except for DENV-2, which retained moderate neutralization (FRμNT50: 37 nM). Similarly, monovalent K8b-F(ab) showed reduced binding to all VLPs with a significant drop in binding to DENV-3, DENV-4, JEV, and ZIKV VLPs. These data show that K8b could occupy quaternary epitopes on the virion surface and confirm that antibody geometry impacts the observed antigen-dependent pattern of binding and neutralization potency of K8b, explaining its potent and broad neutralizing activities.

**Heterogeneous MBC recall responses after mD2VLP stimulation in JEV VLP-primed mice.** To explore the B cell immune responses in vivo mimicking the sequential exposures of JEV and DENV in human infection, we conducted a prime-boost immunization study wherein mice were immunized with mD2VLP or JEV VLP antigens and evaluated the post-vaccination antibody response for neutralization against homologous and heterologous flavivirus serocomplexes (Fig. 6a). Although mice from groups 3-5 which received the heterologous VLPs showed moderate (FRμNT50 GMT: 10-200) polyclonal responses against the four DENV serotypes and JEV, the ZIKV-CR neutralizing activities of these animals were primarily low (FRμNT50 GMT: 10-60) and not significant. Notably, homologous JEV VLP prime-boost immunization consistently induced relatively higher neutralization titers not only to the homologous JEV (FRμNT50 GMT = 164) but also to the heterologous flavivirus serocomplexes not encountered by the vaccinated mice (FRμNT50 GMT, DENV-1 = 115; DENV-2 = 605; DENV-3 = 961, DENV-4 = 603) and recorded the highest anti-ZIKV titers among all groups (FRμNT50 GMT = 64).

To further investigate the MBC clones established by homologous JEV VLP prime-boost immunization, two of the most responsive (FRμNT50 > 120) mice in this group were selected for generating hybridomas using homologous or heterologous antigens, JEV VLP or mD2VLP, respectively, to stimulate MBC proliferation followed by immediate fusion with murine myeloma cells (Fig. 6b). Enzyme-linked immunosorbent assay (ELISA) binding activity showed that polyclonal

hybridomas generated using JEV VLP for stimulation were all JEV-specific (Fig. 6c, left). Seventy-one percent of the 52 monoclones isolated from the 3D2 polyclone secreted JEV-neutralizing mAbs with varying levels of potency at IC50 as strong, moderate, and weak (<1, 1–10, and >10 μg/mL, respectively) (Fig. 6c, right). On the contrary, polyclonal hybridomas generated from the same homologous JEV VLP prime-boost mice but stimulated by heterologous mD2VLP antigens secreted antibodies CR to JEV, DENV, and ZIKV VLPs (Fig. 6d). Fifty-eight percent of the 36 monoclones isolated from the 5D5 polyclone secreted cross-neutralizing mAbs against all six flaviviruses, including JEV, ZIKV, and DENV-1 to -4, while the rest secreted murine mAbs only cross-neutralizing sub-group (Fig. 6e). Most of these mAbs exhibited moderate to weak cross-neutralizing activities (Fig. 6f). Notably, all 52 mAbs (or 59%) from the JEV VLP stimulation strategy belong to the IgG1 isotype, whereas 36 mAbs (or 41%) from mD2VLP stimulation all belong to the IgG3 isotype (Fig. 6g), demonstrating that the heterologous VLP vaccination model induced the IgG3-focused recall responses. Thus, these findings suggest that heterogeneous MBC populations recognizing different flaviviruses can be established from JEV prM/E VLP antigens and recalled upon subsequent heterologous VLP antigen exposure, resulting in IgG isotype-dependent responses with varying neutralizing activities.

Given the IgG3-induced CR response in mice after heterologous antigen stimulation, we next examined if the cross-neutralization activities against ZIKV previously observed in polyclonal human sera were associated with IgG3-specific antibodies by performing a total human IgG3 GAC ELISA and antigen-specific IgG3 capture ELISA. Eighteen of the 31 individuals (58%) with ZIKV VLP-reactive IgG responses showed ZIKV VLP-binding IgG3 antibodies (Supplementary Fig. 7a) despite the similar total human IgG ($P = 0.2024$) and relative IgG3-specific antibodies ($P = 0.1385$) among the healthy controls and DF patients (Supplementary Fig. 7b). Notably, we found a significant and positive correlation between the ZIKV VLP-binding IgG3 subclass and ZIKV neutralization titers (Pearson's correlation, $r = 0.4$; Wilcoxon matched-pairs signed rank test, $W = 0$, $P < 0.05$; Supplementary Fig. 7c). We hypothesized that the low neutralization activity of K8b-IgG1 against DENV-3 observed in this study could be enhanced when expressed as IgG3. Thus, to explore the possible contribution of IgG3 in broad neutralization, one of the broadly neutralizing antibodies, K8b, was expressed as IgG3 (Supplementary Fig. 7d). The purity of these preparations was assessed by SDS-PAGE under reducing and nonreducing conditions showing the expectedly 10-kDa

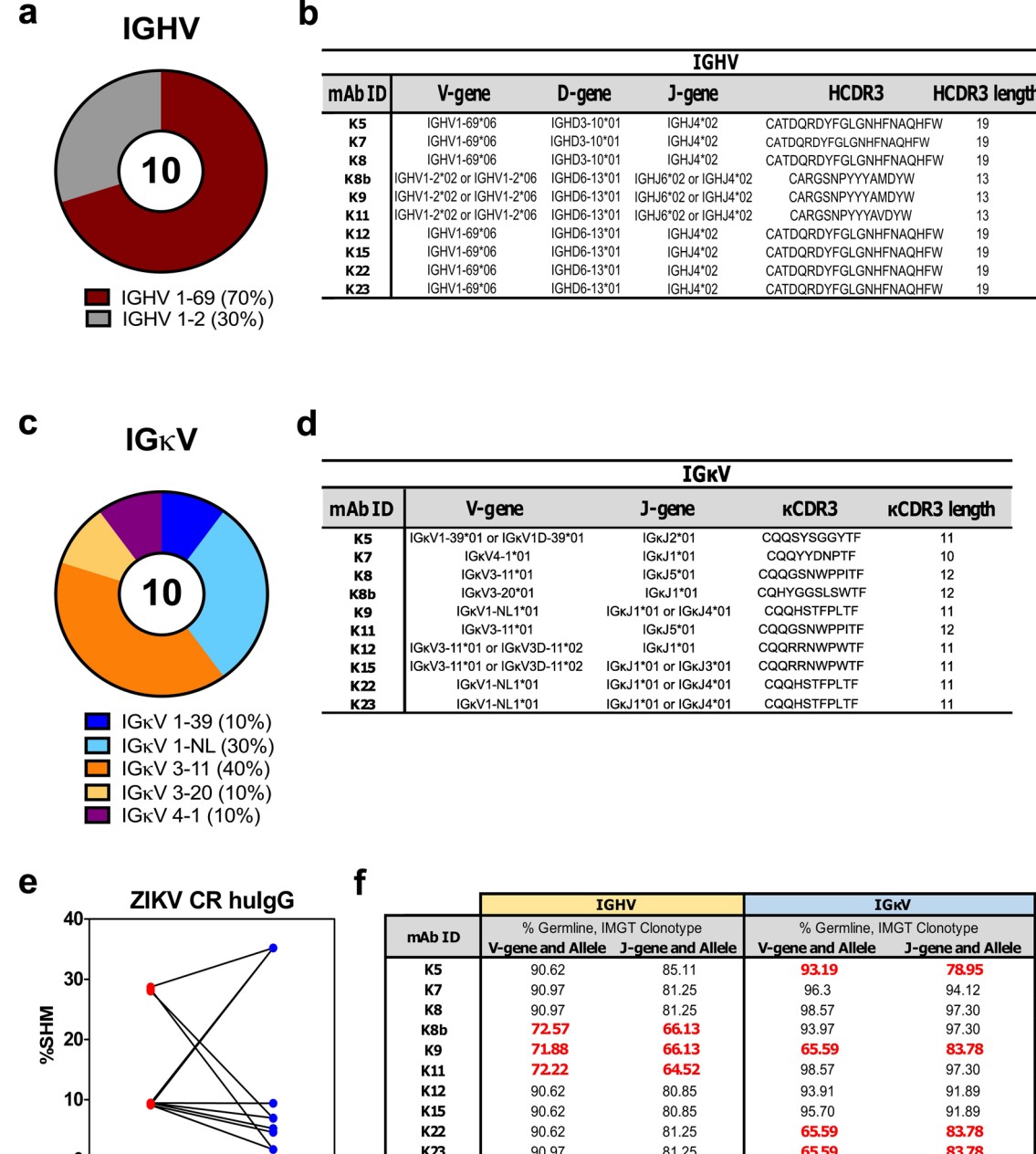

**Fig. 3 Defining the human Ig genes and properties of broadly neutralizing ZIKV-CR huMAbs. a, c** Pie plots showing immunoglobulin heavy (**a**) and kappa light chain (**c**) gene family utilization and distribution of ZIKV-CR neutralizing huMAbs isolated from KH1891. The proportion of IGHV and IGκV gene usage is reported beside the different color schemes. **b, d** Gene family assignments for the 10 purified IgG1 B cell clones from KH1891. IMGT was used to assign the germline reference sequence for IGHV (**b**) and IGκV (**d**) and the relative similarity with the germline clonotype expressed in %. **e** Relative percentage of somatic hypermutation (%SHM) in the paired IGHV and IGκV genes of ZIKV-CR huMAbs. **f** Comparison of the V-J genes and alleles of heavy and light chains of ZIKV-CR huMAbs. Values highlighted in red show increased SHM (>20%) relative to the germline gene family. ZIKV Zika virus.

higher molecular weight of K8b-IgG3 than K8b-IgG1 (Supplementary Fig. 7e). Though K8b-IgG3 showed slightly lower neutralization potency against DENV-3 compared with K8b-IgG1 ($P < 0.05$), no differences were observed in neutralization activities between K8b isotypes against the other prototype strains of DENV-1, -2 and -4, JEV, and ZIKV, as shown by the statistically similar geometric mean IC50 neutralization titers for all virions (Supplementary Fig. 7f–i, $P > 0.05$), suggesting that an exchange of the Fc region contributed minimally in the neutralization potency of K8b. We re-evaluated the natural

immunoglobulin isotype of K8b from the original ZIKV-reactive B cell clone, where it was isolated using human Fc region-specific primers, and confirmed that K8b naturally existed as an IgG1 molecule (Supplementary Fig. 7j, k).

## Discussion
A few studies investigated the interplay of flavivirus immunity in the context of JEV, which is endemic in Southeast Asia; however, these studies focused on animal immunization or ex vivo stimulation using PBMCs from JEV-experienced donors[27,28].

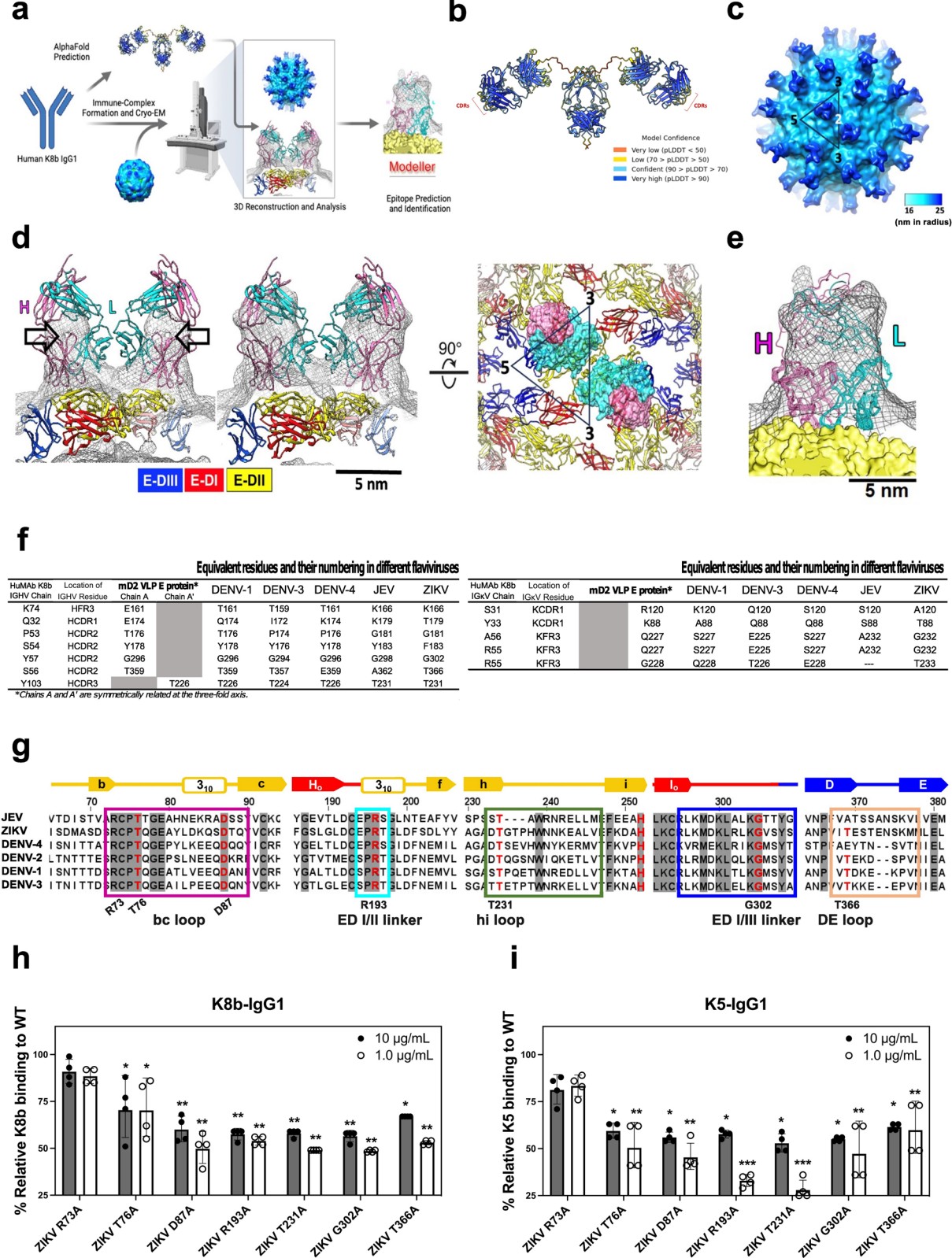

Contrary to the local ZIKV transmission in neighboring Asian countries[29,30], Taiwan, situated in East Asia, detects only imported ZIKV cases without any local transmission. To exclude the possibility of subclinical silent transmission of ZIKV, the seroprevalence studies conducted in southern Taiwan support very few or no local transmission of ZIKV[11,12,31]; thus, providing a unique cohort setting to investigate CR antibody profile after sequential exposures to JEV and DENV. Our study provided serological evidence of a robust anti-ZIKV antibody response elicited by JEV-primed individuals who encountered a

**Fig. 4 Mapping of target flavivirus epitopes of huMAb K8b. a** Structure-based strategy and workflow for the epitope mapping of K8b using mature DENV-2 virus-like particles (mD2VLP). AlphaFold was used to predict the three-dimensional (3D) structure of huMAb K8b expressed as IgG1. K8b-IgG1 was utilized to form an immune complex with mD2VLPs and 3D reconstructed by cryo-EM. The derived structure was further improved by MODELLER to predict the key epitopes and build epitope models. The antibody and apparatus icons were sourced icons and created with biorender.com. **b** AlphaFold model structure for K8b-IgG1 with coloration based on model confidence. The complementarity determining regions (CDRs) were indicated. **c** The cryo-EM structure of K8b-IgG1 and mD2VLP complex revealed the arrangement of E glycoprotein at $T = 1$ icosahedral symmetry. In the configuration, 60 F(ab) molecules (depicted in blue) were bound to 60 envelope proteins on the VLP surface (shown in cyan), while the Fc region was not resolved in the structure. **d** Fitting the predicted structure of K8b-IgG1 into cryo-EM density maps is presented, with contour levels of 3.7 σ (left) and 1.2 σ (middle). Two perspectives are shown: side view (**d**, left and middle) and top view (**d**, right). The separation of variable domains from the constant domain was clearly defined in the density map at higher contour levels, as indicated by the arrows. The heavy and light chains of K8b-IgG1 were shown in purple and cyan, respectively. Heavy and light chain pairs of K8b-IgG1 bound to E proteins at the inter-dimer surfaces. Densities of F(ab) pairs of K8b-IgG1 were shaded in gray. The residues on the E protein interacting with the heavy and light chains of F(ab) were shown by purple and cyan spheres, respectively. Envelope domains, E-DI, E-DII, and E-DIII, are colored in red, yellow, and blue, respectively. The triangle represents an asymmetric unit. **e** A flexible refinement of immune complex structure into cryo-EM map showing ten conformers of K8b-IgG1 predicted by MODELLER. The heavy (H) and light (L) chain variable domains are represented by pink and cyan ribbons, respectively. The mD2VLP is shown as a yellow surface, and the EM map (0.1 σ) is represented as black meshes. The cross-correlation function (CCF) is 0.72. For (**d**) and (**e**), a scale bar of 5 nm is shown. **f** Interacting or contact residues between the E proteins on an mD2VLP and the heavy or light chain of K8b-IgG1, including the equivalent residues (and their AA positions) in other flaviviruses, predicted by MODELLER. **g** A representative amino acid sequence alignment of the E proteins among prototype flaviviruses including selected DENV serotypes, JEV, and ZIKV using Clustal Omega and visualized by JalView 2.11.1.4. Potential binding sites of K8b are highlighted in boxes: bc loop (magenta), ED I/II linker (cyan), hi loop (green), ED I/III linker (blue), and DE loop (flesh). Residues in bold and in red are surface-exposed (solvent-accessible surface area, QSASA > 0.30). Sequence identities were normalized by aligned length, while the positions of the amino acids in E were labeled according to ZIKV PRVABC59. The shaded residues show 100% conservancy among the selected flavivirus genera. Above the sequences, the arrows indicate β-strands, rectangles represent the helices, and lines show the spanning loops and strands in the E structure. The E domains I, II, and III are represented by arrows and lines in red, yellow, and blue, respectively. **h, i** Binding ELISA reactivity profile of purified K8b-IgG1 (**h**) and K5-IgG1 (**i**) at 10 and 1.0 μg/mL antibody concentrations against seven single-site mutants relative to the wild-type ZIKV VLP in ELISA. The consensus amino acid residues in the bc loop (ZIKV R73, T76, and D87), ED I/II linker (ZIKV R193), hi loop (ZIKV T231), ED I/III linker (ZIKV G302), and ED III DE loop (ZIKV T366) were mutated to alanine. For (**h, i**), the statistical differences were determined by two-way ANOVA with Tukey's multiple comparisons test relative to the binding reactivity to the ZIKV WT VLP, and are defined as significant at: *$P < 0.05$, **$P < 0.001$, and ***$P < 0.0001$. The binding reactivity ELISA data shown in (**h, i**) are representative of four independent experiments, expressed as geometric means ± SD. DENV dengue virus, JEV Japanese encephalitis virus, ZIKV Zika virus, ELISA enzyme-linked immunosorbent assay, FRμNT focus-reduction microneutralization test, ANOVA analysis of variance, SD standard deviation.

subsequent natural DENV infection. These serological findings were further confirmed by the isolation of CR huMAbs and heterologous antigen stimulation of the mice primed by JEV antigens. Previous studies showed that sequential exposures to heterologous DENV serotypes generated potent broadly neutralizing antibodies with longevity to ZIKV[7,18,19], reduced ZIKV symptoms in a pediatric cohort in Nicaragua[32,33] and lowered the risk of ZIKV infections in a large prospective Brazilian cohort[34]. While these studies highlighted the importance of infection history on flavivirus immunity[35], they were mainly conducted in countries without reported JEV circulation. Our study demonstrated a promising proof of concept for generating broadly neutralizing antibodies after sequential JEV and DENV exposures.

Our current knowledge of cross-immunity to ZIKV is mainly derived from infection of DENV[9] or other flaviviruses such as WNV[36] and YFV[37] and is limited to the investigation of the interrelationships of two flavivirus serocomplexes[38]. The higher ZIKV-neutralizing titers we observed among the older dengue-infected individuals in our study could suggest that pre-existing MBCs acquired from natural JEV exposure were long-lived and more persistent than those derived from vaccination. Alternatively, it could be due to the repeated exposures to geographically co-circulating flaviviruses over a lifetime in the older age group. In prior age-stratified seroprevalence studies, the accumulation of multitypic dengue and JEV immunity increased with age[39,40], with dengue-immune adults (≥30 years old) showing reduced susceptibility to secondary infections than monotypically exposed children and adolescents under 15 years[39,41]. We cannot completely rule out the possibility of weakly neutralizing antibodies being linked to disease severity among individuals with prior secondary DENV exposures[42–44]. However, passive transfer of cross-neutralizing antibodies from

JEV-vaccinated mice failed to induce antibody-dependent enhancement (ADE) in vitro[45] or affect ZIKV infection or pathogenesis in mice, although these antibodies were low or undetectable and highly cross-reactive[46]. Finally, the high endemicity and geographic co-circulation of mosquito-borne flaviviruses in Asia, such as DENV and JEV, could confer apparent cross-protection provided by B and T cell cross-immunity[27,45,47], thus partly explaining the silent ZIKV transmission or low ZIKV incidences in the region. Similarly, the lack of WNV outbreaks in Latin America was also hypothesized to be due to the co-circulation of related flaviviruses endemic in the Americas[48].

Factors governing the germinal center reaction, affinity maturation, and the induction of circulating MBCs and long-lived plasma cells are complex and remain an active area of research[49]. Previous studies showed that JEV vaccination induces long-lived and protective neutralizing antibodies and memory cytotoxic T lymphocytes in mice[46] and children from 1 year[50] to 5 years[51] after initial JEV administration. One-time passive transfer of anti-JEV-neutralizing antibodies was also more protective against homologous challenge than heterologous viruses[46,52]; however, repeated homologous JEV exposures trigger an extensive MBC response to generate neutralizing antibodies and increased levels of JEV-primed CD8+ T cells[45,53], thus, protecting JEV-infected mice from lethality[54]. In this study, homologous JEV VLP prime-boost immunization effectively induced homologous and heterologous in vitro responses, which were sufficient to form clonally diverse MBCs[54], waiting to be recalled for clonal expansion after DENV exposure. Similarly, the mice study with sequential WNV and JEV exposures suggests that flavivirus-specific MBCs bypass the germinal center in recall response, whose activity depends on the initial clonal diversity of MBCs derived from the initially encountered antigens[55]. Other factors, such as the interval of priming and boosting, the order of

**Table 1 Cryo-EM data collection, refinement, and validation statistics.**

| | mD2VLP-K8b-IgG1 immunocomplex (EMDB-36408) |
|---|---|
| *Data collection and processing* | |
| Magnification | 15,000 |
| Voltage (kV) | 200 |
| Electron exposure (e–/Å$^2$) | 32 |
| Defocus range (μm) | 2.3–4.2 |
| Pixel size (Å) | 4 |
| Symmetry imposed | icos |
| Initial particle images (*n*) | 2035 |
| Final particle images (*n*) | 1246 |
| Map resolution (Å) | 16.3 |
| FSC threshold | 0.143 |
| Map resolution range (Å) | 16.0–19.9 |
| *Refinement* | |
| Initial model used (PDB code) | n.a. |
| Model resolution (Å) | n.a. |
| FSC threshold | |
| Model resolution range (Å) | n.a. |
| Map sharpening *B* factor (Å$^2$) | n.a. |
| Model composition | n.a. |
| Non-hydrogen atoms | |
| Protein residues | |
| Ligands | |
| *B* factors (Å$^2$) | n.a. |
| Protein | |
| Ligand | |
| RMS deviations | n.a. |
| Bond lengths (Å) | |
| Bond angles (°) | |
| Validation | n.a. |
| MolProbity score | |
| Clashscore | |
| Poor rotamers (%) | |
| Ramachandran plot | n.a. |
| Favored (%) | |
| Allowed (%) | |
| Disallowed (%) | |

Refinement was performed by EMAN2 software. The solved cryo-EM structure had suboptimal resolution (16.0–19.9 Å); thus, the requested detailed structural parameters could not be provided with accuracy.
*n.a.* answers were not available.

may not be structurally advantageous in neutralization when expressed as IgG3. Furthermore, the isotypes of the other ZIKV-CR huMAbs from the original B cell clone remain to be identified. The importance of IgG3 in boosting neutralization against various flaviviruses requires future studies, and the potential impact of IgG3 in mediating Fc effector functions observed elsewhere[60,61] will augment our knowledge of the functional relevance of Ig subclasses in the context of heterologous flavivirus infections.

The natural occurrence of huMAbs with high cross-neutralization potency is rare[62], and only a few well-characterized antibodies generated from vaccinations or heterologous infections of humans with prior flavivirus immunity have been described[9,63,64]. Of which, huMAbs MZ20 and MZ54/56 are relevant to the current work since they potently neutralize flaviviruses from three distinct serocomplexes, such as DENV-1 to -4, ZIKV, and JEV or WNV[64]. Other known huMAbs, such as SigN-3C[65], J8/J9[22], or F25.S02[66], were excluded for comparison because only their neutralizing activities against two serocomplexes, DENV and ZIKV, were reported. It is unknown if they could further neutralize a third serocomplex, such as JEV, and might not represent a fair comparison to our huMAbs in the context of our current hypothesis. We initially hypothesized that the potential epitopes of our broadly neutralizing huMAb (bn-huMAb) centered on the conserved regions of flaviviruses from different complexes and spread among the surface accessible loops, such as bc, hi, ij, or fusion loop (FL) (Supplementary Fig. 6a), previously observed in other bn-huMAbs, MZ20[64], MZ54/56[64], and 1C19[67]. On the other hand, K8b and our bn-huMAbs showed lower potency compared to known E DIII lateral ridge-recognizing bn-huMAbs capable of picomolar level-neutralization, such as ZIKV-116[68], 1C11[69], ZK004/006[9], and MZ4[64]. The weaker neutralizing activities against DENV-3 by K8b-IgG1 could be due to a subset of amino acids within the E glycoprotein that is conserved among JEV, ZIKV, and other DENV serotypes but not in DENV-3 (Supplementary Fig. 8a) including, histidine (H158) and proline (P164/166/171) residues, both located at the glycan loop (F$_0$ ß-strand)[47]. The absence of H158 and the change of proline to serine in DENV-3 (Supplementary Fig. 8a, b) possibly affected the glycosylation and conformational landscape-determined epitope accessibility on the surface of DENV, ZIKV, and JEV[70]. Further superimposition of the footprint of K8b on each VLP showed that the electrostatic potential of its light chain on DENV-3 is neutral (Supplementary Fig. 9a–f), indicating lowered binding affinity and, thus, reduced DENV-3 neutralization.

The bn-huMAbs isolated in our study could originate from pre-existing type-specific MBCs that underwent multiple rounds of selection and SHM. Firstly, the B cell repertoire from one donor initially primed with JEV and with recent exposure to DENV involves clonally distinct B cells producing two populations of huMAbs: ZIKV- and the non-ZIKV-CR huMAbs. Among the ZIKV-CR huMAbs, these clonally related antibody sequences showed that the group of antibodies with increased SHMs, such as K5 and K8b, are usually potent neutralizers and complex-specific[5,71,72]. The development of these CR B cells may be driven by clonal selection during the primary encounter with JEV and the subsequent recall and somatic evolution of CR MBC repertoire upon re-exposure to antigenically similar DENV through natural infection[73]. Secondly, our murine MBC stimulation study on JEV VLP-immune mice successfully isolated MAbs recognizing flaviviruses from different serocomplexes, which confirmed that the heterogeneous MBC populations could be established from JEV prM/E VLP antigens and subsequently recalled after encountering a heterologous VLP antigen from a different serocomplex. The reactivation and recall of MBC

heterologous exposures, or the types of antigens used and the duration after the germinal center formation before encountering heterologous antigens could also complicate the outcome of heterologous immunity[56,57]. Further experimental work is needed to determine whether prime-boost immunization of heterologous antigens in mice could also induce similar clonally diverse MBCs in the context of flavivirus infections.

Recent studies in human immunodeficiency virus (HIV) suggested that IgG3 played a vital role in broadly neutralizing antibody responses due to its longer hinge[58,59]. Our findings in the mice immunization study showing that the CR murine monoclonal antibodies isolated from heterologous antigen stimulation are all IgG3 prompted us to examine the IgG3 profile among our DF patients. We further engineered K8b-IgG1 as IgG3 to see if the exchange of Fc region could enhance the moderate neutralizing activity against DENV-3. Despite the significantly positive correlation between ZIKV-VLP-specific IgG3 binding activity and ZIKV-neutralizing antibody titer, no difference in neutralizing activities against the five different flaviviruses was observed between K8b-IgG1 and K8b-IgG3 except DENV-3. As K8b is naturally expressed as IgG1, it

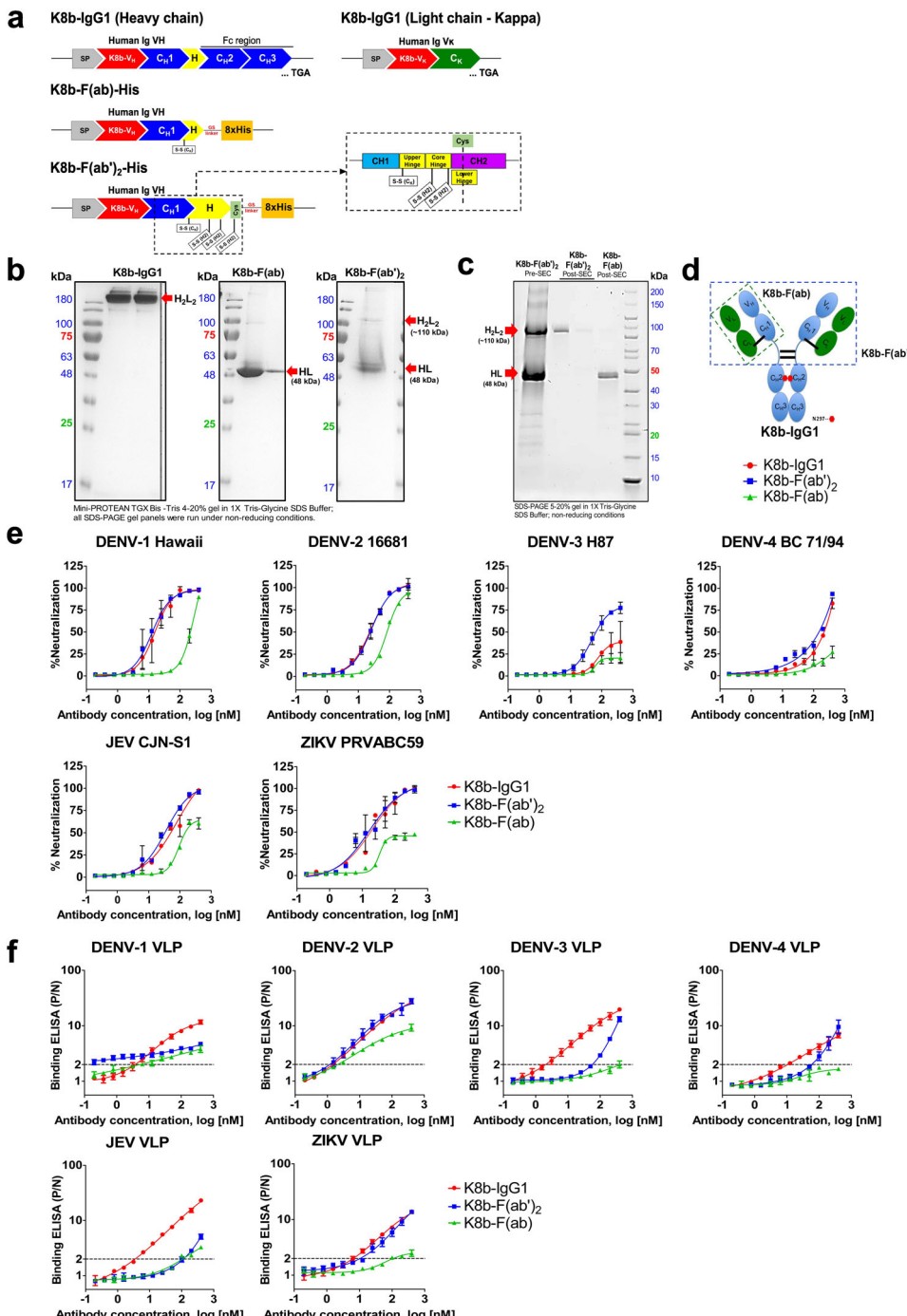

responses upon antigen re-encounter have been initially observed in the T cell immune repertoire[27], highlighting the concerted action of the T cell and B cell immune repertoire in the development and mediation of the broad humoral response after multiple flavivirus exposures. Thirdly, the higher IGHV 1 gene usage in donor KH1891, expressing both the ZIKV- and non-ZIKV-CR huMAbs, is consistent with recurrent lineage usage of VH1/Vκ1 cross-neutralizing DENV-1 and ZIKV[9,10,74]. Despite the similarity in HCDR3 and V(D)J genes among these bn-huMAbs from one donor, the specific target binding is an outcome of unique rearrangements and VL pairing[75] and explains the variability in the observed neutralization. Finally and intriguingly, the isolation of huMAb K5 in our study, which utilized similar heavy (IGHV 1-69) and light chain genes (IGκV 1-39), as

a DENV-3 serotype-specific neutralizing huMAb, 5J7[5]. Both antibodies showed varied HCDR3 sequences and lengths, with K5 showing longer HCDR3 than 5J7 by five residues (Supplementary Fig. 8c, d). Interestingly, amino acid alignment showed that 5J7 had 16 residues that were dissimilar from K5, which spanned across the variable region, suggesting the bn-huMAbs could originate from a serotype-specific B cell that underwent the classical recombination event and then acquired further SHM and insertions at the HCDR3 through selection. All these events indicate that the ZIKV-CR huMAbs we isolated in this study originated from two distinct B cell lineages, meaning separate pathways of MBC development after heterologous antigen stimulation, supporting the importance of flavivirus exposure history on the host humoral response.

**Fig. 5 Generation and functional characterization of purified K8b-IgG1, K8b-F(ab')$_2$, and K8b-F(ab).** **a** Schematic representation of the designed gene constructs coding for full-length human IgG1, F(ab')$_2$, and F(ab) encoding the Ig variable heavy chain genes of K8b (K8b-V$_H$), and hence termed as pVCHIg-hG1-K8b, pVCHIg-K8b-F(ab')$_2$-His and pVCHIg-K8b-F(ab)-His, respectively. The schematic diagram of the expression vector, pVCLIg-hκ, encoding the variable Ig kappa light chain domain of K8b (K8b-Vκ) is also shown in the upper right panel. Each of the molecules was expressed via co-transfection using the kappa light chain expressing plasmid, pVCLIg-hκ-K8b. Inset shows the detailed schematic map of the human Igγ constant domains 1 (C$_H$1) and 2 (C$_H$2), including the IgG1 hinge regions and the actual location where cysteine (Cys) was introduced for increased thermal stability. C$_H$1–3, human Igγ constant domains 1–3; Cκ, human Igκ constant domain; 8x-His, poly-Histidine tag; SP, signal peptide; H, hinge; TGA, stop codon are highlighted. **b** SDS-PAGE of purified human monoclonal antibody, K8b, expressed as full-length IgG1, F(ab), and F(ab')$_2$ molecules visualized under nonreducing conditions. Red arrow marks show the dimeric species of K8b-IgG1 at 150 kDa, monovalent K8b-F(ab) at 48 kDa, and the bivalent K8b-F(ab')$_2$ at 110 kDa. The molecular mass of the protein marker is indicated on the left side, lane 1, for each gel panel. **c** Electrophoretic mobility of purified K8b-F(ab')$_2$ after size-exclusion chromatography (SEC) showing the separation of K8b-F(ab')$_2$ and K8b-F(ab) molecules from the pooled pre-SEC K8b-F(ab')$_2$ fractions in lane 1. Lanes 2 and 3 show the non-reduced form of K8b-F(ab')$_2$ post-SEC and lane 4 shows the K8b-F(ab) molecules post-SEC. The molecular mass of the protein marker is indicated on the right side, lane 5. **d** Cartoon representation of K8b IgG1, F(ab), and F(ab')$_2$ molecules. **e** Neutralization profile of K8b expressed as full-length IgG1, F(ab), and F(ab')$_2$. Prototype virus strains of DENV-1 to -4, JEV, and ZIKV were used for the FRμNT. **f** Binding profile of K8b expressed as full-length IgG1, F(ab), and F(ab')$_2$. Each VLP was properly titrated to obtain equimolar concentrations per well before use in antigen-capture ELISAs. In (**e**) and (**f**), each point shows the mean ± SD of data from two independent experiments. A nonlinear curve was generated for all antibody dilution series for all of the assays. DENV dengue virus, JEV Japanese encephalitis virus, ZIKV Zika virus, ELISA enzyme-linked immunosorbent assay, FRμNT focus-reduction microneutralization test, SD standard deviation, SASA solvent-accessible surface area.

The limitations of the current study include (1) the lack of more plasma samples collected from the dengue patients who recovered at the late convalescent phase (greater than 6 months), making it difficult to extend the current findings to assume generalizability among recovered dengue patients; and (2) the structure-based epitope mapping of huMAb K8b using the DENV-2 VLP and not the virion. Although the cryo-EM data showed that K8b likely recognized the E-dimer surfaces[71,72,76,77] with quaternary interdomain epitopes spanning from D87 of domain I, T231 of domain II, G302 of domain I/III linker, and T366 of domain III, we cannot conclude that K8b binds similarly with the virion since the geometry of VLP is smaller ($T = 1$) than the virion. Expressing K8b in monovalent and bivalent formats also showed an antigen-dependent pattern of binding and neutralization and confirmed that antibody geometry is as important a consideration as the core epitope conservation to conferring exceptional breadth of neutralization[78], especially among EDE-targeting antibodies[79]. The exact footprint and neutralization mechanism of K8b warrant further investigation using virions with high-resolution cryo-EM images and X-ray crystallography.

Overall, we demonstrated in humans and mice how the pre-immunity to flaviviruses would shape the humoral response against an antigen from a different serocomplex to which the host has no previous exposure history, resulting in a unique sub-population of antibodies with broad and potent cross-neutralizing activities. These findings not only enhance our understanding of the quality of humoral response induced in humans after sequential exposures to flaviviral infections but also have implications for alternative vaccination strategies, especially in DENV-endemic regions where ZIKV is known to co-circulate geographically.

## Methods

**Ethics statement, patient acquisition, and isolation of human peripheral blood B cells.** The use of human blood specimens was reviewed and approved by the Institutional Review Board (IRB) of Kaohsiung Medical University (IRB No. KMUHIRB-E (II) −20180092). The donors signed a written informed consent before any data collection. All data were anonymized.

All subjects enrolled in this study were from a dengue cohort in 2014–2015, during the two largest DENV-1 and DENV-2 outbreaks in southern Taiwan[31]. The archived febrile patients' plasma samples were obtained from the dengue clinical cohort study of Kaohsiung Medical University Hospital (KMUH) in southern Taiwan through an ongoing study that enrolled subjects with febrile symptoms suspected of dengue viral infection during the acute phase (i.e., 2 weeks post-infection (poi)). The plasma samples comprised 60 dengue-confirmed individuals and 80 dengue-negative controls, also termed healthy in this study, based on the clinical and laboratory diagnosis upon collection. The nationwide JEV pediatric vaccination program in Taiwan has been implemented since 1968[14,15]. To distinguish between individuals immune to JEV due to vaccination or potential natural infection, the study population was divided into two age groups, namely 20–30 years old (post-JEV vaccination program) and 50–70 years old (pre-JEV vaccination program). Laboratory confirmation of DENV infection consisted of quantitative reverse transcription-polymerase chain reaction (qRT-PCR) and virus isolation[31]. Duplicate blood samples at the time of symptom onset were sent to Taiwan Centers for Disease Control for RT-PCR screening for arbovirus infection using flavivirus consensus and chikungunya virus-specific primers. DENV, JEV, and ZIKV infections are notifiable and reportable infectious diseases in Taiwan[13]. All subjects were confirmed with DENV infection and no autochthonous transmission of Zika virus infections has been observed. Epidemiological data were also obtained, including demographics, travel, and vaccination histories. All subjects were not vaccinated against YFV. De-identified serum samples were evaluated for neutralizing antibodies against JEV, DENV-1 to -4, ZIKV, and YFV.

To comprehensively determine the impact of JEV and DENV immunity on the ZIKV-specific antibody response, we further classified the donors into four groups: (1) JEV$^{pos}$DENV$^{high}$, (2) JEV$^{pos}$DENV$^{low}$, (3) JEV$^{neg}$DENV$^{high}$, and (4) JEV$^{neg}$DENV$^{low}$ by dichotomizing those with JEV FRμNT90 titers ≥10 as JEV$^{pos}$ or <10 as JEV$^{neg}$, and those with DENV FRμNT90 titers ≥80 as DENV$^{high}$ or DENV$^{low}$ (FRμNT90 < 80). Previous criteria on serum-neutralizing antibodies (NT50) ≥30 indicate primary dengue infections[80], while the neutralizing antibodies (NT50) among homotypic and heterotypic dengue-infected individuals at the acute phase of infection ranged from 1:40 to <1:1,280[20]. Considering the epidemiology of dengue infections in Taiwan and the more stringent threshold of neutralization (NT90) we previously set, we considered serum neutralizing (NT90) antibodies ≥80 as evidence of durable dengue titers (DENV$^{high}$) or <80 for confirmed yet low dengue titers (DENV$^{low}$). In addition, JEV-infected individuals with neutralizing antibodies ≥10 were considered seropositive and protective[81]. Since most primary or dengue-infected individuals also showed lower cross-neutralization titers against ZIKV than the infecting DENV or

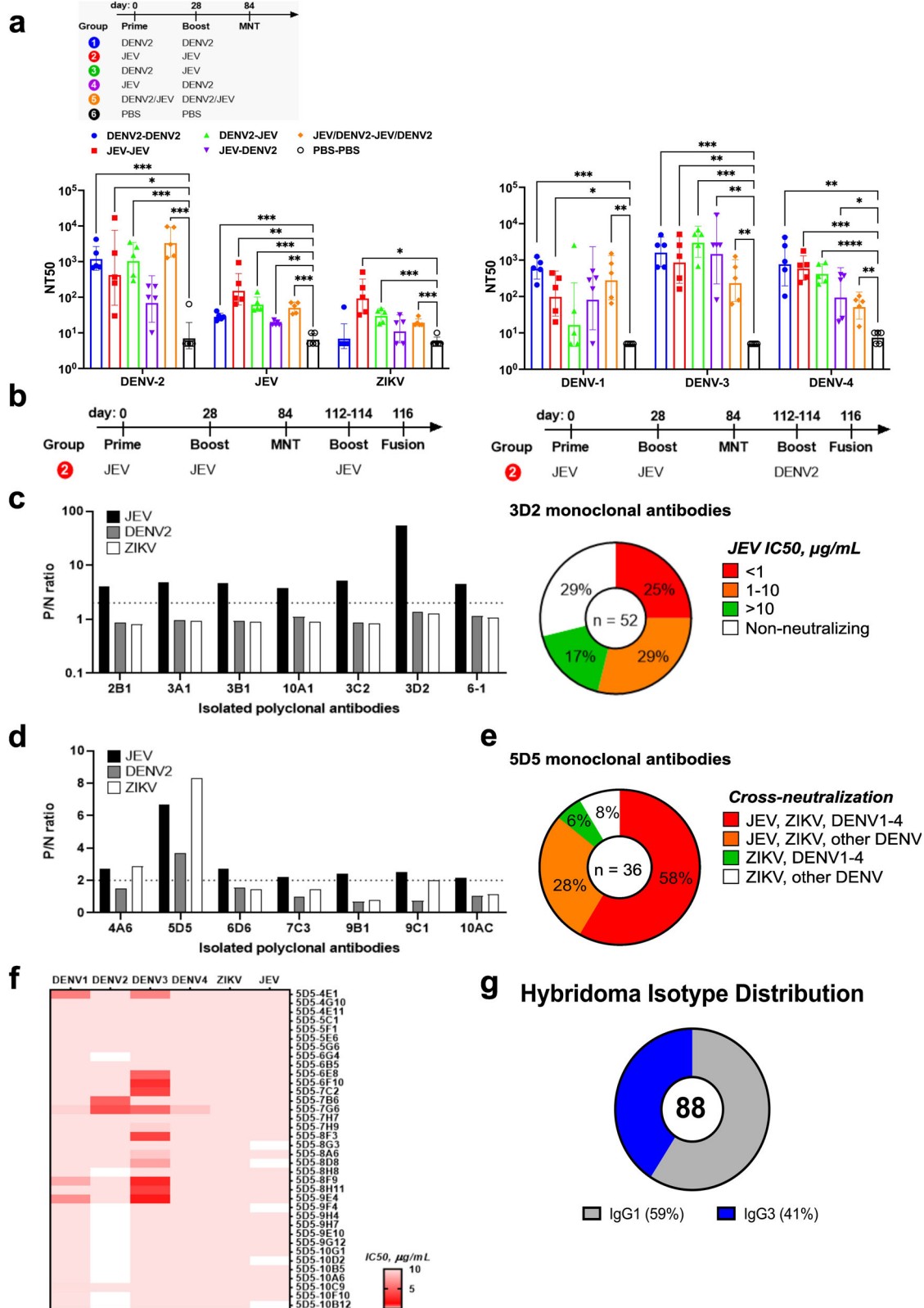

heterologous DENV serotypes[20], we further classified the varying ZIKV neutralization titers from DF individuals as ZIKV[pos] (ZIKV FRµNT90 ≥ 20) and ZIKV[neg] (ZIKV FRµNT90 < 20).

Whole-blood samples were collected into EDTA precoated blood collection tubes (Becton Dickinson) from one of the recalled dengue-immune individuals (KH1891) at late convalescence (>18 months after recovery). Human peripheral blood mononuclear cells (PBMCs) were isolated using the Ficoll-Paque medium (GE Healthcare, Uppsala, SWE) and density gradient method under sterile conditions for sorting.

**Fig. 6 Neutralization antibody profile of murine sera and murine monoclonal antibodies (mAbs) generated from various DENV-2 and JEV VLP prime-boost immunizations. a** BALB/c mice ($n = 5$) were primed and boosted at 0 and 28 days, respectively, with various combinations of mature DENV-2 VLP (mD2VLP) and JEV VLP antigens. Mice were bled at 84 days post-vaccination, and individual serum specimens were evaluated for 50% focus-reduction microneutralization (FRµNT50) activity against DENV, JEV, and ZIKV. The bar plots show GMT ± SD for each group. The $p$ values (*$P < 0.05$, **$P < 0.01$, ***$P < 0.001$, ****$P < 0.0001$) were determined on log-transformed data using one-way ANOVA with Tukey's multiple comparisons post-test. **b** Schedule of boost and myeloma fusion experiments performed in representative mice from the homologous JEV VLP prime-boost immunization. **c, d** Representative mice (JEV FRµNT50 > 120) from JEV prime-boost immunization were further boosted for three consecutive times (112–114 days post-vaccination) with JEV VLP (**c**) or mD2VLP (**d**) and the splenocytes were harvested two days after the last boost and then individually fused with myeloma cells for hybridoma generation. Culture supernatants from selected polyclonal hybridomas were tested for positive VLP ELISA reactivity (P/N ratio ≥ 2) and further subcloned by limiting dilution to isolate monoclonal cells. **c** 71% of the 52 JEV-specific monoclones isolated from 3D2 polyclone secreted JEV-neutralizing mAbs with varying potencies at 50% inhibitory concentration (IC50, µg/mL) of unpurified supernatant. **c** Right: Pie plot showing murine monoclones selected from polyclonal hybridomas and the distribution of 52 type-specific 3D2 mAbs with varying breadth of neutralization. **d** A total of 36 monoclones were isolated from 5D5 polyclone. Percentages represent the proportions of secreted mAbs according to their breadth of cross-neutralization within the different serocomplexes. **e** Pie plot showing murine monoclones selected from polyclonal hybridomas and the distribution of 36 cross-neutralizing 5D5 monoclonal antibodies. The relative proportion of monoclones with varying breadth of neutralization is reported beside the different color schemes. **f** Heatmap of DENV, JEV, and ZIKV IC50 (µg/mL, unpurified supernatant) neutralization titers of the 36 cross-reactive mAbs. Data are representative of two independent experiments. **g** Distribution of immunoglobulin (IgG) isotypes identified from hybridoma screening. DENV dengue virus, JEV Japanese encephalitis virus, ZIKV Zika virus, VLP virus-like particle, ELISA enzyme-linked immunosorbent assay, GMT geometric mean titer, ANOVA analysis of variance, SD standard deviation.

**Cells, antibodies, and virus propagations**. Vero (CRL 1587; ATCC, Manassas, VA, USA) and HEK293T (CRL-3216; ATTC, Manassas, VA, USA) cells were grown at 37 °C with 5% $CO_2$ in Dulbecco's modified Eagle's minimal essential medium (DMEM; Gibco, Grand Island, NY, USA) supplemented with 10% heat-inactivated fetal bovine serum (FBS), 110 mg/L sodium pyruvate, 0.1 mM nonessential amino acids (Gibco, Life Technologies, Grand Island, NY, USA), 2 mM L-glutamine, 20 mL/L 7.5% $NaHCO_3$, 100 IU/mL penicillin and 100 µg/mL streptomycin. All the cells used were free of mycoplasma contamination and checked regularly by following the commercial protocol (InvivoGen, Hong Kong).

Serotype-specific anti-DENV VLP, anti-JEV VLP, and anti-ZIKV VLP mouse hyperimmune ascitic fluid (MHIAF) or sera were either in-house prepared[82] or kindly provided by Dr G.-J. Chang (recently retired from Division of Vector-borne Diseases, Centers for Disease Control and Prevention, DVBD-CDC, Fort Collins, CO, USA).

All flaviviruses were propagated on Vero cells using DMEM supplemented with 2% heat-inactivated FBS, 100 IU/mL penicillin and 100 µg/mL streptomycin for 5–7 days. Infected cell culture supernatants were harvested, clarified by centrifugation at 2000 x g, and stored in aliquots at −80 °C. The virus strains used in this study were DENV-1 Hawaii, DENV-1 BC 245/97, DENV-2 16681, DENV-2 94 Puerto Rico (PR), DENV-3 H87, DENV-3 KH9800235, DENV-4 BC 71/94, DENV-4 H241, JEV CJN-S1, YFV-17D, ZIKV-MR 766, and ZIKV-PRVABC59. All prototype DENV serotypes 1–4, JEV, and ZIKV virus strains were provided by Dr G.-J. Chang (Diagnostic and Reference Laboratory, DVBD-CDC, Fort Collins, CO, USA), while clinical isolate DENV-3 KH9800235 was provided by one of the co-authors, Dr Y.-H. Chen.

**Focus-reduction microneutralization test (FRµNT)**. To measure the in vitro neutralization activity of the human and murine polyclonal antibodies with DENVs, ZIKV, JEV, and YFV, a focus-reduction microneutralization test (FRµNT) in Vero cells was performed[26]. All human and murine polyclonal sera were heat-inactivated at 56 °C for 30 min prior to use in the experiments. Each purified human monoclonal antibody was serially diluted starting at 20 µg/mL. For human polyclonal sera (starting with 10-fold dilution), two-fold serial dilutions were incubated with equimolar concentrations (200 ffu/well) of each of the flavivirus stocks diluted in serum-free DMEM and incubated at 37 °C in 5%

$CO_2$ for 1 h before adding to Vero cells seeded at 180,000 cells/mL the day prior. After 1 h adsorption, the cells were overlaid with 1% carboxymethylcellulose (Sigma-Aldrich Inc., St. Louis, MO, USA) in DMEM with 2% FBS. Immunostaining was performed by adding serotype-specific MHIAF. The infection foci were visualized using a peroxidase substrate kit, Vector VIP SK-4600 (Vector Laboratories, Inc., Burlingame, CA, USA), following the manufacturer's instructions. Each plate included infected and uninfected cell controls and naïve human serum. The micro-neutralization antibody titers of each serum against each virus were defined as the reciprocal of the antibody dilution with the percentage of reduction of virus infectivity relative to the challenge virus dose for each plate. The obtained actual foci count from cells infected with the specific virus (DENV-1 to −4, JEV, or ZIKV) in the absence of plasma were used as virus control, representing 100% infection. We performed the analysis at FRµNT50, 75, and 90 as 50%, 75%, or 90% reduction, respectively, using four-parameter nonlinear regression (version 9.5.1, GraphPad Software Inc.) and found that reporting the more stringent FRµNT90 showed a more conservative threshold of virus neutralization titers and, consequently, a better grasp of exposure history and breadth of neutralization to aid the serum classification as primary or repeat DENV (Supplementary Data 1).

**Focus-reduction microneutralization (FRµNT)-ELISA**. We established an in-house ELISA-based microneutralization test for high-throughput measurement of virus neutralization by detecting the optical density (OD) signal from infected cells. This method was used for virus neutralization screening in hybridoma, small-scale and large-scale expressed and purified human monoclonal antibodies. The standard FRµNT was performed as described above with modifications. Specifically, virus-infected plates were incubated for 72 h at 37 °C with 5% $CO_2$, fixed with ice-cold 75% acetone in 1× PBS for 30 min, and air dried. Each plate included infected and uninfected cell controls and naïve human serum or purified isotype control antibodies. The ELISA was performed by adding 100 µL/well of unpurified, equimolar concentrations of pan-flavivirus anti-E murine MAb FL0231 and anti-NS1 murine MAb mFL0221[83], which were kind gifts from Dr L.-K. Chen (Tzu Chi University Hospital, Hualien, Taiwan). After incubating for 1 h at 37 °C, the plates were washed five times with 1× PBS/0.1% Tween-20 washing buffer. Peroxidase-conjugated goat anti-mouse IgG (Jackson ImmunoResearch

Laboratories Inc., West Grove, PA, USA) diluted at 1:5,000 in 5% skimmed milk-1× PBS (0.1% Tween-20) was added to the wells, and the plates were incubated at 37 °C for 1 h. Finally, the plates were washed ten times, and the bound conjugate was detected with TMB substrate by measuring the absorbance at 450 nm/630 nm. All the optical density (OD) readings were normalized by subtracting the mean OD of uninfected controls from the mean OD of virus controls or the mean OD of samples. The cut-off value (for each plate) for the virus back titration is the mean normalized virus controls as a function of the working virus dilution, determined by applying a three-parameter nonlinear curve fit. The microneutralization antibody titers of each serum or purified monoclonal antibodies against each virus were defined as the reciprocal of the antibody dilution that reduced virus infectivity (and consequently, color development) by 50% (FRμNT50) or 90% (FRμNT90) relative to the infecting virus with control antibody or serum for each plate. The obtained relative optical density (OD) values were normalized to those derived from cells infected with the specific virus (DENV-1 to –4, JEV, or ZIKV) without monoclonal antibodies. The half-maximal inhibitory concentration for monoclonal antibodies (IC50) was determined using four-parameter nonlinear regression (version 9.5.1, GraphPad Software Inc.). All data were log-transformed for analysis, processed, and graphed by GraphPad Prism (version 9.5.1, GraphPad Software Inc.).

**Single B cell sorting, co-cultivation with EL4-B5 cells, and B cell screening by ELISA.** Freshly isolated PBMCs from KH1891 were stained on ice with fluorescently-conjugated anti-human antibodies: CD19-PE-Cy7 (Mat. No. 557835, BD Biosciences Pharmingen™, San Diego, CA, USA), IgM-PE (Code No., 709-116-073, Jackson ImmunoResearch Laboratories Inc., West Grove PA, USA), IgA-APC (Code No., 109-135-011, Jackson ImmunoResearch Laboratories Inc., West Grove, PA, USA), and IgD-FITC (Mat. No., 555778, BD Biosciences Pharmingen™, San Diego, CA, USA) to select CD19+IgM−IgA−IgD− B cell subpopulation (Supplementary Fig. 2b–d). Single cells were sorted according to % yield excluding cell duplets and gated on the live CD19+IgM-IgD-IgA- B cells to determine the target fraction described previously[21]; compensation controls comprised positively stained and negative or unstained cell populations. Sorted B cells were immediately co-cultivated with irradiated EL4-B5 feeder cells expressing CD40L at a seeding density of 3 B cells per well in 384-well plates in the presence of IL-21 and IL-2 in complete Iscove's modified Dulbecco medium (IMDM, Thermo Fisher Scientific, Inc., Rockford, IL, USA) incubated at 37 °C with 5% $CO_2$ and left undisturbed for 2 weeks. After 13 days of cultivation, 40 μL of the culture supernatant from each well was collected and independently screened for binding to the cell culture supernatant containing ZIKV particles by antigen-capture ELISA. ELISA readings were reported according to the average positive-to-negative (P/N) ratio at $A_{450nm/630nm}$ of each sample (reference at 630 nm). The P/N ratio was computed after normalization with the negative control ($NC_{cell}$) obtained from the cell culture supernatant with EL4-B5 cells. Purified total human IgG controls were included per plate and were purified from PV10 (PC) and TW2 ($NC_{TW2}$) plasma by affinity chromatography using 1 mL HiTrap Protein G HP column (GE Healthcare, Uppsala, SWE) and 0.2 μm filtered pH-adjusted buffers. Positively screened B cells were ranked from the highest to lowest ELISA P/N values, which was the basis for prioritizing human monoclonal antibody expression.

**IgG1, IgG3, F(ab), and F(ab')2 plasmid construction, human Ig gene amplification, and expression vector cloning.** In preparation for recombinant immunoglobulin G1 (IgG1) antibody production, in-house expression plasmids pVCHIg-hG1 and pVCLIg-hκ were developed for independent cloning of the human Ig variable heavy (VH) and light (VL) chain genes, respectively. Transcription was controlled by the human cytomegalovirus early gene promoter (PCMV). Both expression plasmids contain the bovine growth hormone poly(A) signal (BGH pA) and the Kanamycin resistance (Kan$^R$) cassette. To facilitate the linearization of the expression vectors, AgeI and SalI were added at the respective 5'- and 3'- ends of the pVCHIg-hG1-vector, while the AgeI and BsiWI sites were introduced at the 5'- and 3'- ends of pVCLIg-hκ expression vectors, respectively (Supplementary Fig. 10a).

To construct the K8b-F(ab) expression plasmids, the full-length K8b VH region with the human constant heavy chain 1 (hCH1) cassette was pre-amplified, including the overlapping sequences encoding the linker and 8x-His residues at the 3'- end. For the F(ab')2 expression vector, the amplification of the insert was performed from the hCH1 cassette until the leucine (L117) residue located at the 5'- end of the hCH2 cassette, which is five nucleotides downstream of the third conserved cysteine (Cys112) residue for heavy chain dimerization. The F(ab) and F(ab')2 expression plasmids were modified to contain an 8x-Histidine tag at the carboxyterminal segment with QuikChange II site-directed mutagenesis kit (Stratagene, La Jolla, CA, USA) (Fig. 5a). An additional cysteine residue (Cys118) was inserted in between the L117 and the Histidine tag sequence to enhance the thermostability of F(ab')2 molecules (Fig. 5a). An inverse PCR was set up to amplify necessary regulatory elements in the pVCHIg-hG1 vector backbone. The resulting PCR fragments shared complementary ends and were joined by subsequent sequence and ligation-independent cloning (SLIC)[84]. PCRs were carried out with the Phusion High Fidelity DNA Polymerase according to the manufacturer's protocol (New England Biolabs, Inc., Ipswich, MA, USA). The generated plasmids were recovered by a standard transformation protocol into chemically competent E. coli. Standard transformation was performed with the addition of pre-warmed SOC media after incubation at 37 °C for 1 h with shaking at 200 rpm, and 100 μL was plated onto Luria-Bertani (LB)-agar Petri dishes containing 100 μg/mL Kanamycin. Sequences were verified by sequencing and herein termed pVCHIg-K8b-F(ab')2-His and pVCHIg-K8b-F(ab)-His. A similar inverse PCR was set up to amplify the variable region expressing K8b and the regulatory elements in the pVCHIg-hG1 vector backbone without the human Ig constant gamma 1 (IgY1) region, while an independent PCR using SLIC-adapted primers was performed to amplify the human Ig Y3 region from pFUSE-CHIg-hG3 (allele 01) including its four hinge domains (InvivoGen, Hong Kong). All the expression plasmids contain the same promoter, enhancer, and selection marker gene cassettes. Primers used for constructing pVCHIg-K8b-IgG1, pVCHIg-K8b-F(ab')2-His, pVCHIg-K8b-F(ab)-His, pVCHIg-K8b-IgG3, and pVCLIg-hK-K8b are summarized in Supplementary Data 2.

Human genes encoding immunoglobulin (Ig) variable heavy (IgHV) and Ig variable kappa (IgκV) chains from ZIKV-reactive B cells were recovered using a two-step RT-PCR strategy. Total RNA was reverse transcribed from lysed B cells, primed using random hexamers (Invitrogen, Carlsbad, CA, USA), and digested with RNAse H at 37 °C for 20 min to remove any excess RNA[84]. Human IgHV and IgκV were amplified independently by two rounds of PCR amplifications using modified primer sets (Supplementary Data 3). The first-round PCR used forward primers in the heavy and light chain functional leader sequence and reverse primers specific to the constant region of IgHV or IgκV, preserving information about the variable region and the isotype. The second-round PCRs were performed with primers

annealing to the 5'- end of the variable (V) genes starting from the framework region 1 (FR1) and their respective reverse primers specific to the joining (J) region of IgHV or IgκV. All primers used in the second-round PCR were designed in preparation for the in vitro assembly using the sequence- and ligation-independent cloning (SLIC) method. SLIC-adapted primers were designed to generate a blunt 5'- end and to contain 20–30 bp sequences complementary to the 5'- or 3'-ends of the expression vector and flanking the human IGHV or IGκV sequences. All PCR reactions were performed with a total volume of 40 μL per sample containing 0.5 μM each of the forward or reverse primers, 300 μM each of the dNTP mix, and 1.25 U One Taq DNA Polymerase (Invitrogen, Carlsbad, CA, USA). All second-round or cloning PCR reactions with gene-specific primers were performed with 3.5 μL of unpurified first PCR products. Each round of PCR was treated as follows: initial denaturation for 30 s at 94 °C, then 50 cycles of denaturation at 94 °C for 30 s, annealing at 58 °C for 30 s, and extension at 72 °C for 55 s (1st PCR) or 45 s (2nd PCR), followed by a final extension of 68 °C for 5 min and a cool down to 4 °C. Each of the successfully amplified Ig variable heavy (IGHV) and Kappa (IGκV) genes was visualized by gel electrophoresis on a 2% (wt/vol) agarose gel in preparation for cloning.

The amplified IGHV and IGκV fragments were assembled in vitro using the SLIC method with the in-house generated pVCHIg-hG1 and pVCLIg-hκ expression vectors, respectively. Briefly, 50 ng of the IGHV and IGκV fragments were independently combined with 100 ng of the respective linearized expression vectors, assembled in vitro as part of a SLIC reaction, and then transformed into chemically competent *E. coli* cells (Supplementary Fig. 10b). Standard transformation was performed followed by spread-plating onto Luria-Bertani (LB)-agar Petri dishes containing 100 μg/mL Kanamycin. Screening of single bacterial colonies containing the IGHV/IGκV inserts was also performed by PCR, and positively screened cultures were grown for 16–18 h at 37 °C with moderate shaking in 5 mL LB (Difco Laboratories) broth containing 100 μg/mL Kanamycin (Sigma-Aldrich Inc., St. Louis, MO, USA). Plasmid DNA was purified with QIAprep Spin columns (Qiagen, Santa Clarita, CA, USA) and sent for sequencing to confirm the insert identity with the original PCR products.

**Human Ig gamma constant region identification**. To determine the natural Ig isotype of the ZIKV-CR huMAbs from donor KH1891, another PCR was set up to amplify the human Ig gamma (Ɣ) constant fragment or Fc region was amplified from freshly prepared cDNAs of the original ZIKV-CR B cells. Universal oligonucleotide primers were designed to target the Fc region spanning domains 1–3 (ƔCH1–3) and the hinge region (Supplementary Data 4). Internal primers targeting ƔCH2 were also designed for use as sequencing primers. All PCR reactions were performed with a total volume of 50 μL per sample containing 0.4 μM each of the forward or reverse primers, 400 μM each of the dNTP mix, and 1.25 U One Taq DNA Polymerase (Invitrogen, Carlsbad, CA, USA). Each round of PCR consisted of initial denaturation for 60 s at 94 °C, then 45 cycles of denaturation at 94 °C for 30 s, annealing at 65 °C for 30 s, and extension at 68 °C for 60 s, followed by a final extension of 68 °C for 5 min and a cool down to 4 °C. Successfully amplified Ig Fc genes, including controls, were visualized by gel electrophoresis on a 2% (wt/vol) agarose gel and sent for sequencing to confirm the identity.

**Human Ig gene sequence analysis**. To define the Ig gene structure, particularly the framework, V(D)J gene assignment, and the complementarity determining region (CDR) boundaries, we analyzed the sequences by comparing the variable gene of the generated IGHV and IGκV fragments with the human Ig sequences using the international ImmunoGenetics (IMGT)/V-Quest germline gene database[85,86]. The closest V(D)J or V-J genes and alleles for the respective heavy and kappa light chains were identified by alignment of the first nucleotide to the conserved second cysteine codon of the V-region in the germline database. Only the sequences with at least 70% V-gene and allele identity with the germline were expressed. Following the IMGT numbering delimitations, analysis of the junction region showed the translated Ig CDR3, which extends from the conserved cysteine (Cys104) to the conserved tryptophan-glycine (Trp-Gly-X-Gly) motif in all human JH gene segments or the conserved phenylalanine-glycine motif (Phe-Gly-X-Gly) in all human Jκ gene segments. Non-silent nucleotide mutations at the V-regions were also generated, which gave rise to distinct amino acid changes. All B cell clones with productive V-genes were expressed. Sequences that were non-productive, out-of-frame, or with premature stop codons were excluded from expression. Sequences matching light or heavy chain sequences from a pseudogene in the germline were also omitted. All the remaining paired sequences with accurate and productive antibody transcripts were expressed as full-length human IgG1 in this study.

Somatic hypermutation (SHM) level (%SHM) was calculated as the divergence of the antibody variable domains from the assigned closest germline sequences at the nucleotide level. Similarly, the amino acid (AA) changes resulting from non-silent nucleotide mutations (%AA change) were evaluated as the divergence in the amino acids relative to the number of germline AA. AA changes were identified following the IMGT physico-chemical classifications based on hydropathy, volume, and IMGT physicochemical properties, and whether the AA change belongs to the same or different class as indicated by (+) or (-), respectively[87,88]. IMGT initially established four types of AA changes: very similar (+++), similar (++-, +-+), dissimilar (--+, -+-, +--), and very dissimilar (---)[88]. We simplified this grouping into two categories: similar or dissimilar AA changes, wherein similar AA changes follow all (+++) or (++-, +-+) any two of the IMGT similarity criteria, while dissimilar AAs belong to only one (--+, -+-, +--) or completely different (---) IMGT classification.

**Antibody expression, IgG quantification, and purification**. Adherent human embryonic kidney (HEK) 293T cells were transfected with equimolar amounts of in-house generated human immunoglobulin G1 (IgG1) heavy and kappa light chain expression vectors using branched polyethylenimine (PEI) (Sigma-Aldrich Inc., St. Louis, MO, USA) at an optimal 1:3 total DNA/PEI ratio. Transient expression of recombinant human IgG antibodies was performed in micro-scale and large-scale cultures for 5 and 7 days, respectively. The collected cell culture supernatants containing the secreted antibodies were clarified by centrifugation at 3000 × g for 10 min, neutralized with phosphate-buffered saline to a final concentration of 1X, passed through 0.20 μm filters (Sartorius, Gottingen, DE), and analyzed for MAb concentration by indirect ELISA using purified recombinant Protein A and Protein G fusion proteins immobilized onto 96-well plates. For large-scale expression, human IgG1 or IgG3 monoclonal antibodies were purified by affinity chromatography using Protein G-coupled agarose beads (Millipore, MA, USA). After binding the supernatant with the beads for 16–18 h at 4 °C, the beads were manually packed onto a 1 mL-polypropylene chromatography column (Qiagen, Santa Clarita, CA, USA) at a flow rate of 1 mL/min, followed by equilibration with 10 column

volumes (CV) of PBS (pH 7.0) washing buffer. The antibody was eluted with 0.1 M Glycine buffer (pH 2.3) onto 96-well plates containing 50 μL 1 M Tris-HCl (pH 8.9) in twenty-four successive fractions of 300 μL. High-antibody fractions were collected per antibody, and each was dialyzed against Tris buffer (pH 6.5) thrice at 4 °C using Slide-A-Lyzer® dialysis cassettes (Thermo Fisher Scientific Inc., Rockford, IL, USA). The purified IgG concentration was estimated using a Qubit® protein assay kit (Life Technologies, Thermo Fisher Scientific Inc., Rockford, IL, USA). For F(ab) and F(ab')$_2$ antibody production, the transformed cells were utilized to grow and express the histidine-tagged K8b-F(ab) and K8b-F(ab')$_2$ antibodies, and the proteins were purified using Ni-NDA affinity chromatography according to the manufacturer's protocol (Chelating Sepharose Fast Flow; GE Healthcare, Uppsala, SWE).

The purity of the full-length K8b-IgG1, K8b-IgG3, K8b-F(ab')$_2$, and K8b-F(ab) antibody preparations was assessed by sodium dodecyl sulfate -polyacrylamide gel electrophoresis (SDS-PAGE). Specifically, purified proteins (1 μg) were mixed with a nonreducing Laemmli sample buffer, resolved on a 4-20% gradient polyacrylamide gel (Mini-Protean TGX; Bio-Rad, Hercules, CA, USA) in Tris-glycine SDS running buffer, and stained with Coomassie blue dye.

**Immunoglobulin G enzyme-linked immunosorbent assay (IgG ELISA)**. An IgG ELISA was used to test the presence and binding reactivity of antigen-specific IgGs among the dengue-negative or healthy and dengue-infected polyclonal sera or in the cell-culture supernatants co-cultivated with B cells or of the purified human monoclonal antibodies using a similar antigen-capture ELISA protocol with minor modifications[26,83]. Briefly, flat-bottom 96-well MaxiSorp NUNC-Immuno plates (NUNC, Roskilde, DK) were coated with 50 μL of serotype-specific rabbit anti-DENV virus-like particles (VLP) sera (1:500), rabbit anti-JEV VLP sera (1:3,200) or rabbit anti-ZIKV VLP (1:8,000) sera at indicated dilutions in bicarbonate buffer (0.015 M Na$_2$CO$_3$, 0.035 NaHCO$_3$, pH 9.6), incubated overnight at 4 °C, and blocked with 200 μL of 5% milk in 1× PBS with 0.1% Tween-20 (5% milk/ PBST) for 1 h at 37 °C. Equimolar amounts of DENV, JEV, or ZIKV VLPs diluted in the blocking buffer were added to each well, incubated for 2 h at 37 °C, and washed five times with 200 μL of 0.1% PBST. The transient expression of each VLP in this study was carried out as described[83] using previously established transcriptionally and translationally optimized eukaryotic cell expression plasmids[89,90]. The amounts and concentrations of the actual VLPs used were equalized in all wells by dilution to ~25 ng VLP per well, as we have previously determined[26,83]. The VLP concentration was determined from the standard curves generated using purified antigens following a sigmoidal dose-response analysis using GraphPad Prism (version 9.5.1, GraphPad Software, Inc., La Jolla, CA, USA). Individual human or mouse polyclonal sera were initially diluted at 1:1,000, titrated two-fold, added into wells in duplicate, and incubated for 1 h at 37 °C. Dengue-negative or healthy controls, pre-vaccination mouse sera, untransfected HEK293T cell culture supernatant, or IgG isotype control (in-house generated SARS-CoV-2 virus-specific huMAb) were used as negative antibody controls for ELISAs. A similar ELISA protocol was used to determine the binding reactivity of isolated human or murine monoclonal antibodies, initially diluted to 10 μg/mL, followed by two-fold titration. Incubations with the donkey anti-human IgG (1:5,000, Sigma-Aldrich, St. Louis, MO, USA) or goat anti-mouse IgG (1:5,000, Sigma-Aldrich, St. Louis, MO, USA) conjugates and substrate were carried out according to the standard ELISA. The OD450 values, modeled as nonlinear functions of the log10 serum dilutions using a sigmoidal dose-

response (variable slope) equation and endpoint antibody titers from two independent experiments, were determined as the dilutions where the OD value was twice the average OD of the negative control.

As for antigen-specific human IgG and IgG3 detection, the in-house developed antigen-capture ELISA was performed using serum-free grown ZIKV virus-like particles (VLPs) as capture antigens described previously with minor modifications[83]. Polyclonal dengue-infected and healthy control plasma were added to the precoated plates and incubated for 1 h at 37 °C. VLP-reactive total human IgG or IgG3 were detected using HRP-conjugated donkey anti-human IgG or mouse anti-human IgG3 (Invitrogen, Thermo Fisher Scientific, Inc., Rockford, IL, USA) for 1 h at 37 °C at 1:5,000 or 1:1,000, respectively.

The binding of purified K8b-F(ab) and K8b-F(ab')$_2$ molecules was determined alongside purified K8b-IgG1 using the VLP capture ELISA described above. Bound F(ab) and F(ab')$_2$ molecules to the VLPs were detected with 1:40,000 HRP-conjugated goat anti-human F(ab) (Jackson ImmunoResearch Laboratories Inc., West Grove, PA, USA). After washing with PBS ten times, TMB substrate was added into the wells, incubated for 10 min, and the reaction was stopped with 2N H$_2$SO$_4$. Reactions were measured at A$_{450/630}$. The relative binding reactivity of F(ab) and F(ab')$_2$ molecules was determined by comparing the P/N values generated from the replicate wells with that of IgG1 molecules.

**K8b-IgG1 and mD2VLP Immune complex formation and cryo-EM 3D reconstruction**. The previously developed and characterized mature dengue 2 VLPs (mD2VLPs)[26] were generated by transfecting HEK293T cells with the recombinant pVD2i-C18S plasmid using Lipofectamine® 2000 DNA Transfection Reagent (cat. 11668019, Thermo Fisher Scientific, USA) according to the manufacturer's instructions. The culture supernatants were harvested and purified by sucrose gradient centrifugation in TNE buffer (50 mM Tris-HCl, 100 mM NaCl, 0.1 mM EDTA) using a Beckman SW-41 rotor. The fraction with the highest OD reading in ELISA was collected, and the VLPs were resuspended in TNE buffer. These samples were further concentrated using Amicon® Ultra-0.5 centrifugal filter units with 100 kDa cut-off. To form the immune complexes of K8b-IgG1 and mD2VLP, K8b-IgG1 was mixed with purified and concentrated mD2VLPs at a molar ratio of 30 molecules of K8b-IgG1 per mD2VLP particle to prevent inter-spike crosslinking. The complex was incubated at 4 °C overnight. The fresh immune complexes were immediately applied to a glow-discharged Quatifoil 2/2 grid (Quatifoil GmbH, DE) for cryo-EM grid preparation. After removing the excess liquid, the grid was rapidly frozen in liquid nitrogen-cooled liquid ethane using a Gatan CP3. Cryo-EM images were captured at 15,000x magnification with an accelerating voltage of 200 kV using a JEM-2100F transmission electron microscope and a direct electron detector (DE-12 Camera System-Direct Electron, LP) with 6.0 μm pixel pitch (~4 Å at the specimen level). The image processing and 3D reconstruction were performed by a single-particle approach using the EMAN2[91,92] software package. Accordingly, 2,035 spiky and spherical particles were manually selected. Contrast transfer function parameters were estimated internally based on the boxed particles (e2ctf.py). Then, 2D reference-free averaging was performed using the default parameters in EMAN2 and generated an initial de novo model based on the 2D class averages, with icosahedral symmetry imposition. All further refinements were performed with automasking filtering using EMAN2. The final resolution of the structure, incorporating an unresolved Fc fragment, was determined to be 16.3 Å, based on the gold-standard

Fourier shell correlation (FSC) at 0.143. To elucidate the con-joining of the F(ab) fragments, a refinement protocol employing a more permissive masking strategy was implemented, which yielded a structural configuration characterized by the discernible presence of the Fc region juxtaposed with F(ab) fragments exhibiting suboptimal resolution (Supplementary Fig. 5).

**Annotation of antibody CDRs and antibody modeling using AlphaFold2 and MODELLER.** The three-dimensional structure of K8b-IgG1 was determined using the AlphaFold tool in UCSF ChimeraX (version 1.3)[93,94]. The antibody was modeled independently from the antigens using the complete AlphaFold2 pipeline with default parameters[93]. The variable heavy and light chain sequences of K8b have been concatenated. Antibody complementarity-determining regions (CDRs) were annotated with the AHo numbering[95] scheme by the ANARCI program[96]. To evaluate the modeling accuracy of the CDRs, we first superimposed the ±4 flanking amino acid residues of each CDR in predicted models onto native antibody structures. Next, the RMSD of the CDRs was calculated by Prody 2.0[97]. The paratope RMSD was computed similarly after superimposing the paratope of the model onto the native structure.

Simultaneously, the three-dimensional structure of K8b-IgG1 was predicted by comparative modeling using MODELLER (v. 4.0)[98]. The crystal structure of HuMAb P5A-2G9 (PDB ID: 7CZT)[99] was chosen to model the F(ab) domain and the crystal structure of mouse MAb IgG2a (PDB ID: 1IGT)[100] to model the Fc domain and the glycan chains. E proteins and K8b-IgG1 F(ab) domains were fitted rigidly using the Fit-in-Map tool in UCSF Chimera[94] with the maximized CCC (Cross-Correlation Coefficient) score. The symmetry restraints were applied to avoid clashes between neighboring protein molecules. Due to its symmetrical nature, the F(ab) domain would have two-fold rotation-related binding poses, which cannot be distinguished under a low-resolution map. To overcome this ambiguity, we used the protein docking software HEX[101] to calculate the binding energy and determine the most probable binding orientation of the F(ab) domain with another molecule. To further refine the immune complex structure and optimize the antigen-antibody interface, we conducted the flexible fitting procedure using the Flex-EM module in MODELLER[102]. The secondary structure elements in E protein and F(ab) were treated as rigid bodies during the fitting. To reduce the computation expense, only half the size of the F(ab) and three E dimers were included in the molecular dynamics simulation. We performed 20 independent simulated annealing molecular dynamics runs and picked ten final conformers with lower binding free energies for subsequent analysis. The antigen-antibody interfaces were analyzed using PDBePISA[103] and PDBsum[104]. The structure figures in this article were created using Chimera[94] or PyMOL (v. 2.5.4).

**SDM and epitope mapping experiments.** To validate the predicted binding site of K8b-IgG1 on mD2VLP, we used the ZIKV VLP by focusing on the amino acid residues that are: (1) conserved among ZIKV, JEV, and the four serotypes of DENV based on sequence alignment (Supplementary Fig. 6a); (2) located on loops with enhanced solvent accessibility based on SASA results (Supplementary Fig. 6b); and (3) candidate epitope from previously reported broadly neutralizing antibodies[64,105,106]. The previously published and transcriptionally optimized eukaryotic cell expression plasmid transiently expressing the envelope (prM/E) proteins from ZIKV was used in this study[107]. Three amino acid residues in the bc loop region of the ZIKV envelope (E) protein, namely arginine (R73), threonine (T76), and aspartic

acid (D87), were separately mutated to alanine (A). Single-point mutations were also introduced in other loops that span the ZIKV envelope protein, such as the fusion loop (FL) (N103), fg loop (R193), hi loop (T231), $I_OA$ loop or ED I/III linker (G302), and the EDIII DE loop (T366). We also incorporated residues 101 and 103 at the fusion loop for site-directed mutagenesis (SDM) since the binding footprint of the EDE antibodies centered on residue 101 of E protein[108,109].

The mutations in the ZIKV E were generated by overlapping PCR with the appropriate pairs of complementary forward (F) and reverse (R) primers (Supplementary Data 5) according to the manufacturer (QuikChange™ II site-directed mutagenesis kit, Stratagene, La Jolla, CA, USA). Briefly, the PCR reaction of 50 μL contained 200 ng of template, 0.5 μM primer pair, 200 μM dNTPs, and 1 Unit of Pfu Ultra II Fusion HS DNA Polymerase. The PCR cycles were initiated at 95 °C for 2 min to denature the template DNA, followed by 18 cycle amplification cycles, with each cycle consisting of 95 °C for 30s, $T_{m\ no}$ −5 °C for 30s, and 72 °C for 8 min. The PCR cycles were finished with an extension step at 72 °C for 8 min. All PCR products were treated with 5 Units of DpnI at 37 °C for 2 h, followed by visualization on agarose gel electrophoresis. An aliquot of the PCR products was transformed into E. coli competent cells using the standard transformation protocol. The transformed cells were spread on a Luria-Bertani (LB) plate containing 50 μg/mL of Ampicillin (Cyrusbioscience Inc., New Taipei, TW), incubated overnight at 37 °C, and isolated for plasmid DNA. All desired mutations were confirmed by DNA Sanger sequencing. The resulting pCDNA3-ZIKV plasmids expressing the full-length envelope protein with single-site mutations, R73A, T76A, D87A, N103A, R193A, T231A, G302A, and T366A were named after the mutation. Following transient expression on HEK293T, the wild-type and mutant ZIKV VLP-expressing cell culture supernatant were collected, titrated for concentrations, and used for binding IgG-ELISAs described above. Most mutant ZIKV VLPs were expressed at 60~100% efficiency except for ZIKV VLP N103A, which only showed 2% VLP expression (Supplementary Fig. 6c), indicative of a disrupted VLP formation; hence, its exclusion in binding ELISA and further analyses.

**Solvent accessibility estimation of E protein in the VLP structure.** EasyModeller (v. 4.0) was used to align the E protein sequences from DENV-1, DENV-3, DENV-4, ZIKV, and JEV onto the DENV-2 E protein structure in the previously published mD2VLP structure[26]. After generating the monomeric E protein structure using MODELLER[98], an in-house generated PyMOL script was used to transfer the $T = 1$ symmetry of mD2VLP to the E protein, and to generate the VLP structures of the five viruses. The derived VLP structures were subjected to calculate the solvent accessibility (%SASA, the fractional solvent-accessible surface area of the amino acid residue) using POPS[110].

**Sequence analysis of flavivirus E proteins.** Flavivirus E protein sequence alignments were performed using Clustal Omega software with the representative prototype strains and the GenBank accession numbers as follows: DENV-1 Hawaii (KM204119), DENV-2 16681 (KU725663), DENV-3 (KU050695), DENV-4 BC 71/94 (MW945661), JEV CJN-S1 (AY303793), ZIKV PRVABC59 (MK713748), and ZIKV MR766 (NC_012532).

**Mouse experiments.** BALB/c mice were used in all experiments. This study was carried out in compliance with the guidelines for the care and use of laboratory animals of the National Laboratory Animal Center, Taiwan. All animal-use and care procedures were reviewed and approved by the Institutional Animal Care and Use

Committee (IACUC) of the National Chung Hsing University (Approval Number: 106-101R2). All efforts were made to minimize the suffering of the mice. The immunization schedule using different VLP prime-boost strategies is shown in Fig. 6a. Groups of five 3-week-old mice were immunized intramuscularly with 4 µg of each purified JEV VLP and/or mD2VLP in 100 µL of 1× PBS (50 µL in each thigh) at 0 and 28 days. VLPs used in immunizing the mice were not adjuvanted. Mice were bled retro-orbitally at 84 days post-vaccination, and individual serum specimens were evaluated for flavivirus neutralization activity as described above.

**Hybridoma generation and murine mAb screening.** Hybridomas secreting anti-flavivirus monoclonal antibodies were generated from the most responsive JEV VLP-vaccinated mice (JEV FRµNT50 > 120) following the standard protocol[111] with minor modifications[112]. Briefly, JEV VLP-vaccinated mice were boosted intraperitoneally with either 4 µg of purified JEV VLP or mD2VLP emulsified in Freund's incomplete adjuvant (Sigma-Aldrich, St. Louis, MO, USA) for three consecutive days. Two days after the last boost, the splenocytes were harvested and then individually fused with NSI/1-Ag4-1 myeloma cells using 50% polyethylene glycol (PEG) (PEG Hybri-Max™, Sigma-Aldrich, St. Louis, MO, USA). Fused cells were resuspended in RPMI supplemented with 20% FBS, hypoxanthine-aminopterin-thymidine medium, and hybridoma growth factor (Nutridoma™-CS, Roche Diagnostics, Indianapolis, IN, USA). Two weeks post-fusion, supernatants from the selected polyclonal colonies were individually screened for secretion of mAbs by IgG ELISA as described above using JEV, DENV-2, and ZIKV VLPs, and detected with an HRP-conjugated goat anti-mouse IgG. The positive clones (P/N values of ≥2) were subcloned by limiting dilution to isolate monoclonal cells and then further screened for VLP ELISA binding and flavivirus neutralization activities using unpurified supernatants.

**Murine monoclonal antibody isotyping.** Cell culture supernatant containing antibody-secreting hybridomas were analyzed for class and subclass identity, specifically murine IgG1, IgG2a, IgG2b, IgG3, IgA, and IgM, as well as light-chain identity (Kappa or lambda light chains), following the manufacturer's protocol (Pierce® Rapid Mouse Antibody Isotyping Kit, Thermo Fisher Scientific Inc., Rockford, IL, USA). Briefly, the culture supernatant was diluted to a final concentration of 1:100 using Tris-buffered saline (20 mM Tris; 150 mM NaCl). Fifty microliters (50 µL) of the diluted antibody mixture and 50 µL of HRP-conjugated goat anti-mouse IgG+IgA+IgM antibodies were simultaneously added to each of the ELISA strip-wells precoated with six different anti-mouse heavy-chain or two light-chain capture antibodies. The samples were mixed by gently tapping the plate, incubated for 1 h at room temperature, and washed with the proprietary wash buffer. Finally, adding TMB substrate followed by 2N $H_2SO_4$ stop solution revealed the antibody isotype based on which wells in the strip produced color. Wells with the highest response (or darkest color) based on visual inspection or spectrophotometer readout at 450 nm/630 nm indicate isotype and light chain compositions.

**Statistics and reproducibility.** All data were represented as means ± standard deviations (SD) and were analyzed using GraphPad Prism (version 9.5.1, GraphPad Software Inc.). Statistical significance was determined using a one-way analysis of variance (ANOVA) followed by Tukey's multiple comparisons post-test to analyze the ELISA and neutralization titers across multiple groups. The correlation between IgG3 and neutralization titers was calculated using Pearson's correlation and the Wilcoxon matched-pairs signed rank test. $P$ values less than 0.05 were considered significant.

**Reporting summary.** Further information on research design is available in the Nature Portfolio Reporting Summary linked to this article.

## Data availability

All raw data associated with this study are available in the supplementary data or upon request. The source data behind the graphs in the paper are available in Supplementary Data 6. Individual-level epidemiological data are not publicly available due to participant confidentiality and in accordance with the IRB-approved protocol under which the study was conducted. Uncropped and unedited SDS-PAGE and agarose gel images are provided in the supplementary information (Supplementary Figs. 11–14). The cryo-EM density map of mD2VLP-K8b-IgG1 immunocomplex has been deposited to the Electron Microscopy Data Bank under accession number EMD-36408.

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

## Acknowledgements

The authors would like to express their gratitude to the Core Facility Center of National Cheng Kung University for providing access to the JEM-2100F instrument for data collection. The authors also thank the ASGC (Academia Sinica Grid-computing Center) Distributed Cloud resources for their computational support, which is supported by Academia Sinica. Selected icons in Fig. 4 and Supplementary Fig. 2b were sourced icons created with biorender.com. We also thank Dr Félix A. Rey (Structural Virology Unit, Virology Department, Pasteur Institute, Paris, France) for the helpful comments on the manuscript and ongoing follow-up experiments on defining the footprint of huMAb K8b using X-ray crystallography in his lab. This work was partly supported by the Ministry of Science and Technology (MOST), Taiwan (Grant No. MOST-110-2313-B-005-041). Any opinions, findings, and conclusions expressed in this material are those of the authors and do not necessarily reflect the views of MOST.

## Author contributions

G.M.S. wrote the original draft, performed and designed most of the wet lab investigation, co-conceptualized the study design, data generation, data analysis, and data curation. J.U.G. and C.-S.S. designed and led the in vivo investigation, data generation, data analysis, and data curation and helped write, review, and edit the manuscript. S.-R.W. performed the cryo-EM, structural data visualization and formal analysis and contributed to the writing, review, and editing. J.-H.L. performed the structural data modeling, provided the software, formal analysis and structural data visualization in this study, and contributed to the writing, review, and editing of the manuscript. Y.-H.C., W.-H.W., and S.-F.W. were responsible for patient acquisition at the referral hospital, initial epidemiological and molecular diagnostics of donor volunteers, and transportation of the plasma samples to co-investigators. F.-C.C. assisted in the site-directed mutagenesis and structural data visualization. A.B.A. assisted in the data curation and investigation involving K8b-IgG3. G.-W.C. assisted in the acquisition of cryo-EM data. C.-Y.W. provided the feeder cells used in the study and supervised G.M.S. during her research visit in Singapore. D.-Y.C. conceptualized the overall study design, coordinated with Y.-H.C. on the human sample acquisition, supervised and acquired the funding, and contributed to interpreting the results and revising the manuscript.

## Competing interests

The authors declare no competing interests.
