## [Peer Review File · Communications Biology]

This manuscript has been previously reviewed at another Nature Portfolio journal. The manuscript was considered suitable for publication without further review at Communications Biology.

REVIEWERS' COMMENTS:

Reviewer #3 (Remarks to the Author):

Salem et al. describe the isolation of neutralizing antibodies against ZIKV from DENV-infected patients with prior exposure to JEV. Concerns regarding some of the data generated using cryo-electron microscopy have now been addressed through revisions including supplemental panels in which representative micrographs, 2D class averages, and FSC curves are shown. The authors have also improved how the data are presented in figures and have adequately addressed all additional minor points.

Reviewer #4 (Remarks to the Author):

Please find below the review of the rebuttal to reviewers #1 and #2 for the manuscript "Broadly neutralizing antibodies against Zika virus from dengue patients with prior exposure to Japanese encephalitis virus"

Overall the authors have provided thorough responses to address the comments made by reviewers 1 and 2 and have gone to considerable effort to improve the manuscript. I believe the authors have largely adequately address the concerns of the reviewers and the manuscript is suitable for publication. A few minor outstanding comments are given below.

Reviewer 1

2. Another strong claim is that "ZIKV-specific IgG3-mediated cross-reactive neutralization response in humans, which is consistent with the results from the mice immunization study" (lines 257-258). To directly support this claim, the authors would need to compare neutralizing activity of the antibodies expressed as IgG1 vs. IgG3. Otherwise, it is speculative and should be presented as such.

We thank the reviewer for pointing this out. We performed follow-up experiments, whose results are reported as Supplementary Fig. 7. We also included a brief explanation of these results at the Discussion (lines 349-370).

Lines 349-370

"Recent studies in human immunodeficiency virus (HIV) suggested that IgG3 played a vital role in broadly neutralizing antibody responses due to its longer hinge^{55,56}. With our mice immunization study findings that the CR murine monoclonal antibodies isolated from heterologous antigen stimulation are all IgG3, we further examined the IgG3 profile among our DF patients. Our results suggested that 18 of the 31 individuals (58%) with ZIKV VLP-reactive IgG responses showed ZIKV VLP-binding IgG3 antibodies (Supplementary Fig. 7a) despite the similar total human IgG ($P=0.2024$) and relative IgG3-specific antibodies ($P=0.1385$) among the healthy controls and DF patients (Supplementary Fig. 7b). Importantly, a significant and positive correlation with ZIKV neutralization titers was found (Supplementary Fig. 7c). We hypothesized that the low neutralization activity of K8b-IgG1 against DENV-3, observed in this study (Fig. 2g), could be enhanced when expressed as IgG3, which prompted us to express K8b as IgG3 (Supplementary Fig. 7d, e) to explore the possible contribution of IgG3 subclass in broad neutralization. However, no differences were observed in neutralization activities between K8b isotypes against the prototype strains of DENV-1 to -4, JEV, and ZIKV (Supplementary Fig. 7f-i, $P>0.05$), suggesting that an exchange of the Fc region contributed minimally in the neutralization potency of K8b. Using human Fc region-specific primers, re-evaluation of the natural immunoglobulin isotype of K8b from the original ZIKV-reactive B cell clone where it was isolated confirmed that K8b naturally existed as IgG1 (Supplementary Fig. 7j, k). In contrast to the ZIKV-specific, IgG3-mediated cross-neutralizing polyclonal response in humans and mice

immunization study, the isotypes of the other ZIKV-CR huMAbs from the original B cell clone remain to be identified, and the importance of IgG3 in boosting neutralization against various flaviviruses require future studies. "

The authors have gone to considerable effort to address the comment made here and provided novel experimental data which directly addresses the comment.

However, as this is novel experimental data, it should be first mentioned in the results section of the manuscript along with a figure call out.

4. The actual IC50/FRNT90 values of antibodies should be included as a supplement. Currently, only heatmaps are shown, making it difficult to assess potency.

We agree that providing the actual IC50/FR μ NT50 values of the antibodies allows a better assessment of potency. In this lieu, we revised the Figure to show the heatmap and the actual IC50/FRuNT50 values, as shown in Figure 2e.

We also provided the exact numerical values on the minimal binding concentration of the antibodies against the VLPs to provide more precise information for better interpretation of results, as shown in Figure 2d.

The revised heatmap with numerical values is clear and addresses the reviewer's comment.

The figure and figure legend are somewhat out of order on the page and are still a little confusing when trying to navigate the figure. Could the authors please try to address this formatting and minimally order the figure legend alphabetically.

Minor weaknesses:

1. Lines 106-107 and Figure 1b: the authors suggest that titers of JEV and ZIKV neutralizing antibodies are higher in older age groups because MBCs acquired from natural infection are superior than those from vaccination. An alternative explanation is that older individuals are more likely to have had multiple exposures to DENV/JEV, boosting cross-reactive responses.

We agree with the reviewer that the older individuals must have been exposed to multiple flaviviruses, including but not limited to DENV or JEV. Although we have no further information on other flaviviruses to which the study population must have been exposed other than those tested in this study, multiple exposures to DENV or JEV in older individuals may have contributed to the more robust cross-reactive responses. We included this alternative explanation in the concluding sentences of the second paragraph of our Discussion, as reflected below:

Lines 316-324:

"...The higher ZIKV neutralizing titers we observed among the older dengue-infected individuals in our study also suggest that pre-existing MBC responses acquired from a potential natural JEV exposure were long-lived and persistently circulated than the MBCs acquired after JEV vaccination. Although we are limited by information on the older individuals' flavivirus exposure history other than those tested in this study, the older age group was also more likely to have had multiple exposures to DENV and JEV or related flaviviruses not limited to DENV or JEV, increasing the quantity and composition of CR antibodies and consequently, boosting their overall polyclonal humoral responses against related flaviviruses.

While the authors have acknowledged the alternative hypothesis proposed by the reviewer, the included text is still somewhat unclear and needs modification, the statement "the older age group was also more likely to have had multiple exposures to DENV and JEV" is also quite concrete without supporting data and while this intuitively makes sense, it may be better being somewhat qualified.

A shorter, revised statement containing something similar to the below could improve readability. The “The higher ZIKV neutralizing titers we observed among the older dengue-infected individuals in our study” could suggest that MBCs acquired from natural JEV exposure are long-lived and more persistent than those derived from vaccination. Alternatively, this effect could be due to the potential for a greater number of exposures to flaviviruses over a lifetime in the older age group.

4. As the authors suggest their findings have implications for future vaccination strategies, the finding that HOMOLOGOUS JEV prime-boost vaccination in mice generates higher anti-ZIKV neutralizing antibody titers than heterologous JEV-DENV prime-boost should be discussed in more detail.

We have commented on this phenomenon in the Discussion and referred to studies that might explain more robust cross-neutralizing response induced by homologous JEV exposures than the heterologous JEV exposures. We have prepared a separate paragraph (third paragraph) in the Discussion, which we hope more clearly and extensively explains the observed results upon repeated JEV exposures:

Lines 333-348:

“Factors governing the germinal center reaction, affinity maturation, and the induction of circulating MBCs and long-lived plasma cells are complex and remain an active area of research (Victora and Nussenzweig, 2012). Previous studies showed that JEV vaccination induces long-lived and protective neutralizing antibodies and memory cytotoxic T lymphocytes in mice (Konishi et al., 1998) and children from one year (Feroldi et al., 2013) to five years (Sohn et al., 2008) after initial JEV administration. Besides the increased levels of JEV-primed CD8+ T cells (Zhang et al., 2020; Chen et al., 2020), repeated homologous JEV exposures trigger an extensive MBC response to generate neutralizing antibodies, thus protecting JEV-infected mice from lethality (Larena et al., 2013). Furthermore, the mice study with sequential heterologous WNV and JEV exposures suggests that flavivirus-specific MBCs bypass the germinal center in recall response, whose activity depends on the initial clonal diversity of MBCs derived from the initially encountered antigens (Wong et al., 2020). Such observations could extend to related antigens such as ZIKV, as shown in the mice immunization results in this study. The heterologous JEV-DENV VLP prime-boost vaccination in mice led to a heterogeneous B-cell population waiting to be reactivated for clonal expansion. The ample supply of antigen in the germinal centers (Larena et al., 2013) allowed the antibodies to evolve from affinity-matured B cell clones continuously (Victora and Nussenzweig, 2012) and consequently induce the potentially neutralizing heterotypic responses.”

While the authors state that the paragraph added to the discussion (above) addresses the reviewers query as to why homologous JEV prime-boost vaccination in mice generates a more robust anti Zika response than heterologous vaccination strategies, from reading the amended paragraph I am struggling to see how this conclusion could be drawn. The paragraph seems to state that both homologous and heterologous prime boost strategies are equally effective at generating potentially neutralizing heterotypic responses. Could the authors please clarify their position on the reviewers comment and amend the included paragraph to improve readability.

5. It’s unclear why/how the authors chose only a subset of those previously described in the field to compare to the newly identified cross-reactive neutralizing antibodies. Other known human antibodies that potentially cross-neutralize DENV1-4 +/- ZIKV that were not discussed include SIgN-3C (PMIDs: 29263863, 28422757, 32348755), J8/J9 (PMID: 31820734), and F25.S02 (PMID: 37090561).

We thank the reviewer for this comment, and for pointing out this issue. Indeed, numerous human monoclonal antibodies (huMAbs) were isolated from flavivirus infections or vaccination with broad neutralization potencies. However, to provide relevance and context to our current hypothesis, we focused on bn-huMAbs induced by vaccination or sequential infections of humans with prior flavivirus immunity and found only a few papers (Dussupt et al., 2020, PMID: 32015557; Robbiani et al., 2017,

PMID: 28475892; Rogers et al., 2017, PMID: 28821561). We expanded the selection criteria to include human antibodies based on their (1) similarity with the breadth of neutralization with K8b, (2) similarity in the target epitopes or binding sites with K8b; and (3) degree or extent of published information on the structural basis of broad neutralization. HuMAb MZ4 (Dussupt et al., 2020) elicited broad and potent neutralization against medically-important mosquito-borne flaviviruses belonging to three different serocomplexes, such as the four DENV serotypes, ZIKV and JEV, similar to K8b. We revised the fifth paragraph of our Discussion completely and included humAb MZ54/56 (Dussupt et al., 2020), which has a potent neutralization against DENVs, ZIKV, and WNV, and referred to it in the second paragraph of the Discussion:

Lines 347-351:

“The natural occurrence of huMAbs with high cross-neutralization potency is rare (Dejnirattisai et al., 2015), and only a few well-characterized antibodies generated from vaccinations or heterologous infections of humans with prior flavivirus immunity have been described (Rogers et al., 2017; Dussupt et al., 2020; Robbiani et al., 2017). Of which, huMAbs MZ20 and MZ54/56 are relevant to the current work since they potently neutralize flaviviruses from three distinct serocomplexes, such as DENV-1 to -4, ZIKV, and JEV or WNV (Dussupt et al., 2020).”

Other bn-huMAbs, MZ20 (Dussupt et al., 2020, PMID: 32015557) and 1C19 (Smith et al., 2013, PMID: 24255124) were also chosen because they recognize residues similar to the putative binding sites of K8b such as the bc loop (R73) or the ED I/III linker (G302) in the case of MZ4.

The authors inclusion of this section provides a rationale for their selection and goes some way to addressing the reviewers comment. Perhaps they could also include a short section/sentence on the antibodies they have excluded along with rationale.

Reviewer 2

Remarks to the author

The complex serology of flaviviruses has been described in multiple studies. The concept that primary exposure to one viral species influences the antibody response following secondary exposure to an antigenically related virus is not novel nor a phenomenon unique to flaviviruses. This has been discussed extensively in the influenza and SARS-CoV-2 literature. While this paper describes some nice experiments, the major conclusions presented by the authors (lines 327-330) are not a conceptual advance.

While I believe the reviewers comment is valid, the authors here study a specific flavivirus infection scenario and demonstrate how this exposure history influences antibody generation and responses. This study adds considerable detail and nuance to a generally accepted concept.

6. The suggestion that k8 binds VLPs bivalently is of interest. The work does not present the cryo-data well, including the methodology. For example, the methods element does not describe how (and in what molar ratios) the antibody is bound to the VLP before application on the grid. The resolution is very low, and the heavy chain is unresolved.

I believe the modified presentation of the CryoEM data and the methodology is adequate and the major concerns of the reviewer have been addressed. While the structure was solved at a modest resolution, the maps provided unambiguously show bound antibody Fv domains and the modelling to position the full antibody is plausible.

We thank the Reviewers for their time in carefully reviewing our manuscript and for the thorough assessment of our work. We also greatly appreciate the recruitment of Reviewer #4, who provided unbiased and constructive feedback on how we can further improve the manuscript and prepare it for final submission.

Having reviewed all the comments, we addressed the remaining issues raised by Reviewer #4 point-by-point on the rebuttals made by the previous round of reviewers, as outlined and described below. All the necessary revisions were highlighted in yellow in the mark-up version of the manuscript accordingly.

Reviewer 1

2. Another strong claim is that “ZIKV-specific IgG3-mediated cross-reactive neutralization response in humans, which is consistent with the results from the mice immunization study” (lines 257-258). To directly support this claim, the authors would need to compare neutralizing activity of the antibodies expressed as IgG1 vs. IgG3. Otherwise, it is speculative and should be presented as such.

The authors have gone to considerable effort to address the comment made here and provided novel experimental data which directly addresses the comment.

However, as this is novel experimental data, it should be first mentioned in the results section of the manuscript along with a figure call out.

- We are delighted about the generally positive feedback from the reviewer on our efforts to address the previous comments on the impact of the IgG3 subclass in the cross-neutralizing responses. We appreciate the reviewer’s thorough feedback and agree to have this novel experimental data reported and mentioned first in the Results section (Lines 275-295).

Lines 275-295:

“Given the IgG3-induced CR response in mice after heterologous antigen stimulation, we next examined if the cross-neutralization activities against ZIKV previously observed in polyclonal human sera were associated with IgG3-specific antibodies by performing a total human IgG3 GAC ELISA and antigen-specific IgG3 capture ELISA. Eighteen of the 31 individuals (58%) with ZIKV VLP-reactive IgG responses showed ZIKV VLP-binding IgG3 antibodies (Supplementary Fig. 7a) despite the similar total human IgG ($P=0.2024$) and relative IgG3-specific antibodies ($P=0.1385$) among the healthy controls and DF patients (Supplementary Fig. 7b). Notably, we found a significant and positive correlation between the ZIKV VLP-binding IgG3 subclass and ZIKV neutralization titers (Pearson’s correlation, $r=0.4$; Wilcoxon matched-pairs signed rank test, $W=0$, $P<0.05$; Supplementary Fig. 7c). We hypothesized that the low neutralization activity of K8b-IgG1 against DENV-3 observed in this study (Fig. 2g) could be enhanced when expressed as IgG3. Thus, to explore the possible contribution of IgG3 in broad neutralization, one of the broadly neutralizing antibodies, K8b, was expressed as IgG3 (Supplementary Fig. 7d). The purity of these preparations was assessed by SDS-PAGE under reducing and non-reducing conditions showing the expectedly 10-kDa higher molecular weight of K8b-IgG3 than K8b-IgG1 (Supplementary Fig. 7e). Though K8b-IgG3 showed slightly lower neutralization potency against DENV-3 compared with K8b-IgG1 ($P<0.05$), no differences were observed in neutralization activities between K8b isotypes against the other prototype strains of DENV-1, -2 and -4, JEV, and ZIKV, as shown by the statistically similar geometric mean IC50 neutralization titers for all virions (Supplementary Fig. 7f-i, $P>0.05$), suggesting that an exchange of the Fc region contributed minimally in the neutralization potency of K8b. We re-evaluated the natural immunoglobulin isotype of K8b from the original ZIKV-reactive B cell clone, where it

was isolated using human Fc region-specific primers, and confirmed that K8b naturally existed as an IgG1 molecule (Supplementary Fig. 7j, k).”

We have further elaborated on the results in the Discussion section. We also included experiments based on prior work that examined the impact of Fc-mediated functions on IgG3 (Richardson et al., 2019, PMID: 31841557; Onodera et al., 2021, PMID: 34508662), which are shown in Lines 351-364. We hope we have adequately addressed any remaining issues on these experimental concepts.

Lines 351-364:

“Recent studies in human immunodeficiency virus (HIV) suggested that IgG3 played a vital role in broadly neutralizing antibody responses due to its longer hinge (Moyo-Gwete et al., 2022; Vidarsson et al., 2014). Our findings in the mice immunization study showing that the CR murine monoclonal antibodies isolated from heterologous antigen stimulation are all IgG3 prompted us to examine the IgG3 profile among our DF patients. We further engineered K8b-IgG1 as IgG3 to see if the exchange of Fc-region could enhance the moderate neutralizing activity against DENV-3. Despite the significantly positive correlation between ZIKV-VLP-specific IgG3 binding activity and ZIKV neutralizing antibody titer, no difference in neutralizing activities against different flaviviruses was observed between K8b-IgG1 and K8b-IgG3, except DENV-3. As K8b is naturally expressed as IgG1, it may not be structurally advantageous in neutralization when expressed as IgG3. Furthermore, the isotypes of the other ZIKV-CR huMAbs from the original B cell clone remain to be identified. The importance of IgG3 in boosting neutralization against various flaviviruses requires future studies, and the potential impact of IgG3 in mediating Fc effector functions observed elsewhere (Richardson et al., 2019; Onodera et al. 2021) will augment our knowledge of the functional relevance of Ig subclasses in the context of heterologous flavivirus infections.”

4. The actual IC50/FRNT90 values of antibodies should be included as a supplement. Currently, only heatmaps are shown, making it difficult to assess potency.

The revised heatmap with numerical values is clear and addresses the reviewer’s comment.

The figure and figure legend are somewhat out of order on the page and are still a little confusing when trying to navigate the figure. Could the authors please try to address this formatting and minimally order the figure legend alphabetically?

- We thank the reviewer for the positive feedback on the revised heatmap with numerical values. We also appreciate the reviewer’s insightful suggestions on the remaining details of this display figure. We have made the necessary adjustments to the corresponding figure legends for the binding ELISA and neutralization graphs. The legends have also been repositioned to appear immediately after each panel set to improve navigation and referencing. Additionally, the legends have been reformatted alphabetically for better organization. The revised heatmaps with numerical values were also repositioned immediately after each panel for continuous readability. We hope these changes have adequately addressed any confusion concerning the formatting and readability of Figure 2.

In addition, we have also ensured that the citation and numbering of the display figures and supplementary items in the manuscript were in a chronological manner in which they appear. Repeated mentioning of the display or supplementary figures was also minimized throughout the manuscript to avoid confusions.

Minor weaknesses:

- 1. Lines 106-107 and Figure 1b: the authors suggest that titers of JEV and ZIKV neutralizing antibodies are higher in older age groups because MBCs acquired from natural infection are superior than those from vaccination. An alternative explanation is that older individuals are more likely to have had multiple exposures to DENV/JEV, boosting cross-reactive responses.**

While the authors have acknowledged the alternative hypothesis proposed by the reviewer, the included text is still somewhat unclear and needs modification, the statement “the older age group was also more likely to have had multiple exposures to DENV and JEV” is also quite concrete without supporting data and while this intuitively makes sense, it may be better being somewhat qualified.

A shorter, revised statement containing something similar to the below could improve readability.

the “The higher ZIKV neutralizing titers we observed among the older dengue-infected individuals in our study” could suggest that MBCs acquired from natural JEV exposure are long-lived and more persistent than those derived from vaccination. Alternatively, this effect could be due to the potential for a greater number of exposures to flaviviruses over a lifetime in the older age group.

- We value the reviewer’s feedback to improve the brevity of our work. Following the recommendation of the reviewer, a more concise version of the original statement is now presented in the Discussions section (Lines 317-321):

Lines 316-320:

“The higher ZIKV neutralizing titers we observed among the older dengue-infected individuals in our study could suggest that pre-existing MBCs acquired from natural JEV exposure were long-lived and more persistent than those derived from vaccination. Alternatively, it could be due to the repeated exposures to geographically co-circulating flaviviruses over a lifetime in the older age group.”

We also thank the reviewer for suggesting to qualify better the alternative hypothesis above concerning age-related robust immunity. To provide a more thorough explanation, we highlighted two prior age-stratified seroprevalence studies conducted in dengue-infected individuals emphasizing the higher incidences of individuals with multi-typic immunity with age and more importantly, how this results in reduced susceptibility to other dengue infections (Castanha et al., 2013, PMID: 22800513; Rodriguez-Barraquer et al., 2011; PMID: 21245922). We hope that by citing these prior works, we offered a better understanding and qualified (or contextualized) the concept of robust immunity with age.

Lines 320-323:

“In prior age-stratified seroprevalence studies, the accumulation of multitypic dengue and JEV immunity increased with age (He et al., 2020; Castanha et al., 2013), with dengue-immune adults (≥ 30 years old) showing reduced susceptibility to secondary infections than monotypically exposed children and adolescents under 15 years (Castanha et al., 2013; Rodriguez-Barraquer et al., 2011).”

- 4. As the authors suggest their findings have implications for future vaccination strategies, the finding that HOMOLOGOUS JEV prime-boost vaccination in mice generates higher anti-ZIKV neutralizing antibody titers than heterologous JEV-DENV prime-boost should be discussed in more detail.**

While the authors state that the paragraph added to the discussion (above) addresses the reviewers query as to why homologous JEV prime-boost vaccination in mice generates a more robust anti Zika response than heterologous vaccination strategies, from reading the amended paragraph I am struggling to see how this conclusion could be drawn. The paragraph seems to state that both homologous and heterologous prime boost strategies are equally effective at generating potentially neutralizing heterotypic responses. Could the authors please clarify their position on the reviewers comment and amend the included paragraph to improve readability.

➤ We value the reviewer's feedback on this part of the Discussion.

Previous studies show that JEV exposures induce long-lasting humoral and cellular immunity in animals and vaccinated humans (Konishi et al., 1998, PMID: 9573260; Feroldi et al., 2013, PMID: 23442823; Sohn et al., 2008, PMID: 18294743), with repeated homologous JEV exposures establishing an extensive and protective MBC repertoire (Zhang et al., 2020, PMID: 36659231; Chen et al., 2020, PMID: 32501510; Larena et al., 2013; PMID: 23388724). Our results showed either homologous or heterologous prime-boost immunization is sufficient to induce *in vitro* cross-neutralization titers; however, we chose only the homologous prime-boosting immunization mice to investigate if the JEV-primed mice is sufficient to form heterogeneous memory B cell clones, which are ready to be recalled upon encountering different flavivirus antigens. Further experimental work is needed to determine whether prime-boost immunization of heterologous antigens in mice could also induce similar clonally diverse MBCs in the context of flavivirus infections. Other factors, such as the interval of priming and boosting, and the order of heterologous exposures, could also complicate the outcome of heterologous immunity. Therefore, further experimental work is needed in the context of flavivirus infections. Nonetheless, we agree with the reviewer that the prior statement in the Discussion needs more clarity on our position on the naturally complex flavivirus immunity, especially in the context of heterologous flavivirus exposures. We have significantly revised the third paragraph of the Discussion, and we hope that the latest revision more clearly defines our position on the results of our prime-boost experiments.

Lines 333-350:

“Factors governing the germinal center reaction, affinity maturation, and the induction of circulating MBCs and long-lived plasma cells are complex and remain an active area of research (Victoria and Nussenzweig, 2012). Previous studies showed that JEV vaccination induces long-lived and protective neutralizing antibodies and memory cytotoxic T lymphocytes in mice (Konishi et al., 1998) and children from one year (Feroldi et al., 2013) to five years (Sohn et al., 2008) after initial JEV administration. One-time passive transfer of anti-JEV neutralizing antibodies was also more protective against homologous challenge than heterologous viruses (Konishi et al., 1998; Bosco-Lauth et al., 2011); however, repeated homologous JEV exposures trigger an extensive MBC response to generate neutralizing antibodies and increased levels of JEV-primed CD8⁺ T cells (Zhang et al., 2020; Chen et al., 2020), thus, protecting JEV-infected mice from lethality (Larena et al., 2013). In this study, homologous JEV VLP prime-boost immunization effectively induced homologous and heterologous *in vitro* cross-neutralizing responses, which were sufficient to form clonally diverse MBCs (Larena et al., 2013), waiting to be recalled for clonal expansion after DENV exposure. Similarly, the mice study with sequential WNV and JEV exposures suggests that flavivirus-specific MBCs bypass the germinal center in recall response, whose activity depends on the initial clonal diversity of MBCs derived from the initially encountered antigens (Wong et al., 2020). Other factors, such as the interval of priming and boosting, the order of heterologous exposures, or the types of antigens used and the duration after the germinal center formation before encountering heterologous antigens could also complicate the outcome of heterologous immunity

(Elliott et al., 2022; Shaw et al., 2022). Further experimental work is needed to determine whether prime-boost immunization of heterologous antigens in mice could also induce similar clonally diverse MBCs in the context of flavivirus infections.”

5. It’s unclear why/how the authors chose only a subset of those previously described in the field to compare to the newly identified cross-reactive neutralizing antibodies. Other known human antibodies that potentially cross-neutralize DENV1-4 +/- ZIKV that were not discussed include SIgN-3C (PMIDs: 29263863, 28422757, 32348755), J8/J9 (PMID: 31820734), and F25.S02 (PMID: 37090561).

The authors inclusion of this section provides a rationale for their selection and goes someway to addressing the reviewers comment. Perhaps they could also include a short section/sentence on the antibodies they have excluded along with rationale.

- We are delighted for the commendable feedback from the reviewer on the inclusion of related broadly neutralizing human antibodies presented in other works. Indeed, numerous human monoclonal antibodies (huMAbs) were isolated from flavivirus infections or vaccination with broad neutralization potencies. Other antibodies, such as SIgN-3C, J8/J9, or F25.S02 might not represent a fair comparison to our huMAbs in the context of our current hypothesis. Following the reviewer’s recommendation, we have included the missing details and provided a brief statement in our Discussion (fifth paragraph) on the rationale for excluding some previously isolated broadly neutralizing antibodies from flavivirus infections, as shown in Lines 367-373:

Lines 367-373:

“Other known huMAbs, such as SigN-3C (Xu et al., 2017), J8/J9 (Durham et al., 2019), or F25.S02 (Lubow et al., 2023), were excluded for comparison because their neutralizing activities were reported only against two serocomplexes, DENV and ZIKV. It is unknown if they could further neutralize a third serocomplex, such as JEV, and might not represent a fair comparison to our huMAbs in the context of our current hypothesis.”

Reviewer 2

Remarks to the author

The complex serology of flaviviruses has been described in multiple studies. The concept that primary exposure to one viral species influences the antibody response following secondary exposure to an antigenically related virus is not novel nor a phenomenon unique to flaviviruses. This has been discussed extensively in the influenza and SARS-CoV-2 literature. While this paper describes some nice experiments, the major conclusions presented by the authors (lines 327-330) are not a conceptual advance.

While I believe the reviewers comment is valid, the authors here study a specific flavivirus infection scenario and demonstrate how this exposure history influences antibody generation and responses. This study adds considerable detail and nuance to a generally accepted concept.

- We greatly appreciate your positive comments about our study and emphasizing the value of our current work in the context of specific flavivirus infection scenarios. We are pleased to hear that our work has added considerable detail and nuance to the concept under consideration.

6. The suggestion that k8 binds VLPs bivalently is of interest. The work does not present the cryo-data well, including the methodology. For example, the methods element does not describe how (and in what molar ratios) the antibody is bound to the VLP before application on the grid. The resolution is very low, and the heavy chain is unresolved.

I believe the modified presentation of the CryoEM data and the methodology is adequate and the major concerns of the reviewer have been addressed. While the structure was solved at a modest resolution, the maps provided unambiguously show bound antibody Fv domains and the modelling to position the full antibody is plausible.

- We thank the reviewer for commenting on the modified presentation of the cryo-EM and the methodology. We are glad to hear that the major concerns have already been addressed.